# Learning to Explore and Exploit with GNNs for Unsupervised Combinatorial Optimization

**Utku Umur Acikalin**[1]
ua45@cornell.edu

**Aaron Ferber**[1]
amf272@cornell.edu

**Carla P. Gomes**[1]
gomes@cs.cornell.edu

[1]Cornell University, Department of Computer Science

## Abstract

Combinatorial optimization (CO) problems are pervasive across various domains, but their NP-hard nature often necessitates problem-specific heuristic algorithms. Recent advancements in deep learning have led to the development of learning-based heuristics, yet these approaches often struggle with limited search capabilities. We introduce Explore-and-Exploit GNN ($X^2$GNN, pronounced x-squared GNN), a novel unsupervised neural framework that combines exploration and exploitation for combinatorial search optimization: i) Exploration - $X^2$GNN generates multiple solutions simultaneously, promoting diversity in the search space; (ii) Exploitation - $X^2$GNN employs neural stochastic iterative refinement to exploit partial existing solutions, guiding the search toward promising regions and helping escape local optima. By balancing exploration and exploitation, $X^2$GNN achieves superior performance and generalization on several graph CO problems including Max Cut, Max Independent Set, and Max Clique. Notably, for large Max Clique problems, $X^2$GNN consistently generates solutions within 1.2% of optimality, while other state-of-the-art learning-based approaches struggle to reach within 22% of optimal. Moreover, $X^2$GNN consistently generates better solutions than Gurobi on large graphs for all three problems under reasonable time budgets. Furthermore, $X^2$GNN exhibits exceptional generalization capabilities. For the Maximum Independent Set problem, $X^2$GNN outperforms state-of-the-art methods even when trained on smaller or out-of-distribution graphs compared to the test set. Our framework offers a more effective and flexible approach to neural combinatorial optimization, addressing a key challenge in the field and providing a promising direction for future research in learning-based heuristics for combinatorial optimization. [1]

## 1 Introduction

Combinatorial optimization (CO) problems aim to find a discrete solution that optimizes an objective function from a discrete set of feasible solutions constrained by specific problem parameters. These optimization problems frequently emerge in commercial, governmental, and scientific contexts, prompting extensive study in fields such as mathematics, computer science, and operations research. Many combinatorial optimization problems are NP-hard, indicating no polynomial-time exact algorithm exists unless P = NP.

Exact algorithms, which achieve optimality by implicitly or explicitly considering all possible solutions, are typically only tractable for small instances due to their exponential worst-case time complexity. Consequently, another body of work focuses on heuristic algorithms that quickly attain high-quality solutions without optimality guarantees. Work here seeks to balance computational time and solution quality, often exploring the search space through multiple different solutions, and exploiting promising ones (Eiben and Schippers, 1998).

Heuristics are often hand-crafted for a specific problem, exploring problem intricacies to achieve peak performance for a given problem distribution, requiring time and domain expertise. The rise of deep

---

[1]Code is available at https://github.com/utkuumur/X2GNN

learning has enabled new learning-based heuristics that can be automatically tuned for performance using data.

However, current approaches are often limited in their search capabilities, often only iteratively improving a single solution, or naively restarting and forgetting previously generated solutions.

In this paper, we introduce **Explore-and-Exploit GNN** ($X^2$**GNN**, pronounced X-squared GNN), a novel unsupervised neural combinatorial optimization framework. $X^2$GNN combines effective exploration of the solution space with intelligent exploitation of partial solutions.

**Our Main Contributions are:**

1. We propose $X^2$**GNN**, which combines exploration and exploitation for combinatorial search optimization: (i) Exploration - $X^2$GNN simultaneously generates multiple coupled solutions, promoting diversity in the search space; (ii) Exploitation - $X^2$GNN employs neural stochastic iterative refinement to exploit partial existing solutions, guiding the search toward promising regions and helping escape local optima.

2. State-of-the-art performance: $X^2$GNN outperforms existing learning-based approaches on benchmark datasets for the maximum cut, maximum independent set, and maximum clique problems. Additionally, $X^2$GNN is competitive with general OR approaches like Gurobi, and problem specific heuristics like KaMIS, offering improved or comparable solution quality at similar time cutoffs.

3. Strong generalization capabilities: $X^2$GNN generalizes to graphs that are out-of-distribution or up to 4 times larger than those seen during training, while still significantly outperforming other learning-based methods trained on the same distribution as the test set.

4. Rigorous Evaluation: We enhance existing benchmark datasets commonly used in the ML for combinatorial optimization community by including strong traditional baselines and evaluating solvers at comparable runtimes. We additionally allow solvers a 30-minute time limit, which is at least 24 times longer than our longest-running model.

## 2 RELATED WORK

The broad intersection between machine learning (ML) and combinatorial optimization (CO) has seen much work with different facets explored in various surveys (Bengio et al., 2021; Kotary et al., 2021; Cappart et al., 2023). State-of-the-art learning-based primal heuristics specifically can be broadly categorized by their training supervision and solution construction methods. Supervised learning approaches use training data composed of problem instances and corresponding solutions derived from existing solvers (Khalil et al., 2016; Selsam et al., 2019; Nair et al., 2020; Sun and Yang, 2023). However, these may face challenges such as the unavailability of high-quality solvers for all problems and poor generalization capabilities across different problem instances (Yehuda et al., 2020). Despite these challenges, recent studies have shown that diffusion-based training can enhance generalization in supervised learning (Sun and Yang (2023)).

Unsupervised learning approaches have also been explored, differing primarily in whether solutions are constructed autoregressively or not. Earlier non-autoregressive models generate a 'soft' solution in a single step, which is then decoded into a final solution using methods ranging from simple greedy (Karalias and Loukas, 2020) decoding to more sophisticated techniques (Min et al., 2022). As Sanokowski et al. (2024) noted, these approaches can be classified as single-step diffusion methods. These models are notably faster and more scalable than their autoregressive counterparts. Sanokowski et al. (2023) suggest that non-autoregressive solution construction may fail to capture essential dependencies among problem variables and they refer to these types of methods as mean-field approximations.

The earlier single-step non-autoregressive methods are outperformed by autoregressive construction governed by MDPs (Sanokowski et al., 2023; Zhang et al., 2023). However, these models are trained using reinforcement learning (RL) and face high computational needs and poor generalization (Sun and Yang, 2023). Additionally, autoregressive construction does not allow modification of fixed decisions, unlike diffusion-based construction where all variables can be altered at each step.

The success of generative diffusion models (Sohl-Dickstein et al., 2015) made it appealing for CO. For diffusion-based CO approaches, noise is sequentially added to the optimal solution obtained from other solvers in the forward process, and the model learns to iteratively remove this noise in the reverse process. Sun and Yang (2023) models the CO problems as a discrete diffusion problem using Bernouilli and Categorical noise. For MIS, they outperform non-autoregressive models but have similar performance to autoregressive models, suffering from long diffusion schedules. Li et al. (2023) follow the same training procedure as Sun and Yang (2023) but employ gradient-guided noising-denosing rounds during inference. This improves upon Sun and Yang (2023); however, long diffusion schedules and reliance on the gradients hinder effectiveness. Sanokowski et al. (2024) trains a diffusion model to sample from the Boltzmann distribution with the probability of sampling a solution being positively correlated to its objective value. They derive an unsupervised loss function using a continuous Lagrangian relaxation, showing that longer generation schedules increase quality.

## 3 PRELIMINARIES

We consider the broad class of combinatorial optimization problems on graphs and instantiate $X^2$GNN for three NP-hard problems on undirected unweighted graphs $G = (V, E)$. As in many real-world scenarios, we consider distributional versions of these problems, where we are asked to train an algorithm on a dataset of instances and then deploy the algorithm on unseen instances.

**Maximum Clique (MC):** A clique in a graph $G$ is a subset of vertices where every two vertices are adjacent. The Maximum Clique problem involves finding the largest clique in $G$.

**Maximum Independent Set (MIS)**: An independent set in a graph $G$ is a subset of vertices, none of which are adjacent. The Maximum Independent Set problem aims to find the largest independent set.

**Maximum Cut (MCut):** For a graph $G = (V, E)$, the Maximum Cut problem seeks to partition the set of nodes $V$ into two subsets $S$ and $V \setminus S$, maximizing the number of edges between $S$ and $V \setminus S$.

Solutions for these problems can be represented by a binary decision for each node, $Y \in \{0, 1\}^{|V|}$, indicating solution inclusion, $Y_i = 1$ if $v_i \in S$ otherwise 0. Additionally, the maximum clique and maximum independent set problems are closely related; a clique $S$ in graph corresponds to an independent set $S$ in its complementary graph (Cormen et al. (2001)). We use this relationship for all approaches and solve MC problems by solving MIS on the corresponding complementary graph.

## 4 $X^2$GNN FRAMEWORK

$X^2$GNN, illustrated in Figure 1, is an iterative framework that explores the search space by simultaneously generating a pool of $K$-Coupled solutions and exploiting promising ones via stochastic refinement.

$K$-Coupled solutions form a group of $K$ solutions that are built collectively, we refer to each group as a $K$-Couple. We model the $K$-Couple using a multilayer graph, copying the original graph $K$ times to represent the $K$ solutions, and adding auxiliary edges between corresponding nodes in different layers. We model the solution values themselves as node features. We then iteratively feed the $K$-Couple into a graph neural network (GNN) which makes a prediction on each node corresponding to a new $K$-Couple. We train the GNN's outputs at each iteration using a combination of an unsupervised optimization loss, a constraint satisfaction loss, and a diversity loss on the $K$-Couple. Importantly, before feeding a $K$-Couple as input to the GNN, we randomly perturb the solution to help escape local optima. Furthermore, we randomly initialize the $K$-Couple.

Formally, given a problem represented by graph $G = (V, E)$, we represent the GNN input of $K$ solutions at iteration $t$ using ${}^tX \in [0, 1]^{K \times |V|}$, with ${}^tX_u^k$ denoting the feature of node $u \in V$ for solution $k \in 1, \ldots, K$ in iteration $t$. We use ${}^t\hat{Y}$ to denote the $K$-Coupled solution generated at iteration $t$. That is, ${}^t\hat{Y} = g_\theta(G, {}^tX)$. Similarly, we use the notation ${}^t\hat{Y}_u^k$ to denote the probability of node $u \in V$ being in solution $k \in 1, \ldots, K$ generated at iteration $t$.

**Solution Generation and Stochastic Iterative Refinement:** We refer to the first iteration ($t = 1$) of $X^2$GNN as construction and the subsequent iterations ($t \geq 2$) as refinement. During construction, we randomly initialize node features ${}^1X_u^k$ to 0.5 with probability $p$ and 0 with probability $1 - p$,

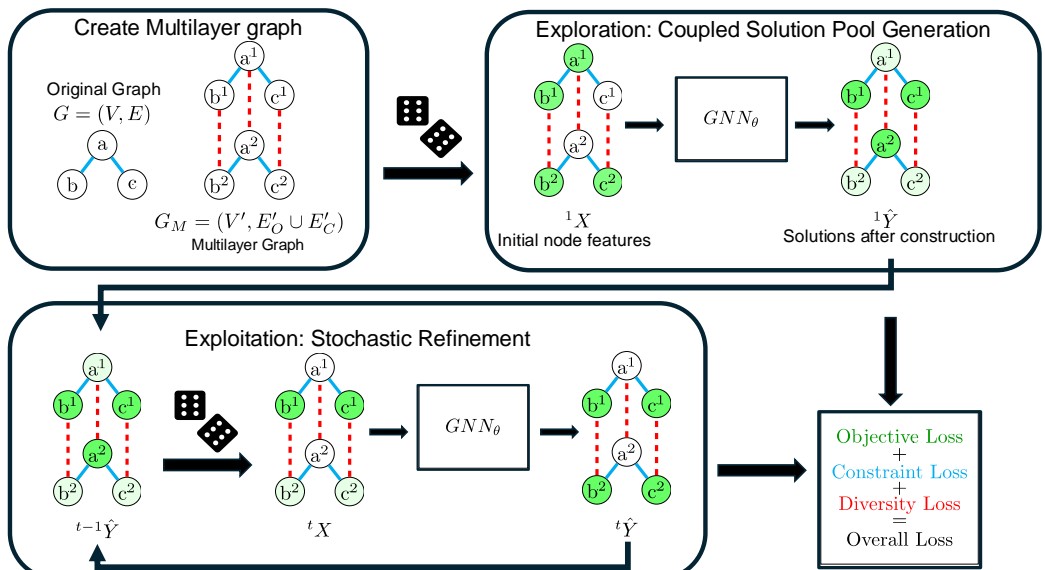

Figure 1: Illustration of $X^2$GNN for a Minimum Independent Set instance. First, a multilayer graph is created from $K = 2$ copies of the original graph, with cross edges to couple solutions. Copies of the original edges ($E'_O$) are drawn in blue, and cross edges ($E'_C$) are drawn in red. Node features correspond to the probability of being in the solution, representing soft solutions. Initially, when generating the coupled solutions, the features are random. These features are fed into a GNN to obtain $K$ soft solutions. During stochastic refinement, the GNN iteratively takes solutions from the previous time step, randomly perturbs them, and generates new solutions. Stochastic refinement can be repeatedly applied at inference and is done once during training. Finally, the training loss is calculated for all generated solutions using the objective value, Lagrangian term, and diversity.

with $p$ being 0.95 in practice. Essentially, we initialize the $K$-Couple with unbiased solutions while introducing diversity to break symmetries for nodes with identical degrees.

During refinement, we randomly set the previous iteration's output $^{t-1}\hat{Y}$ to 0 with probability $\phi$ to generate new node features $^tX$.

The parameter $\phi$ offers a natural way to control exploration and exploitation. If $\phi = 1$, no information from the previous iteration is used, maximizing exploration. If $\phi = 0$, the previously generated solutions are maintained, and the method will deterministically refine the $K$-Couple, maximizing exploitation. Thus, high values of $\phi$ lead to exploration whereas low values lead to exploitation. We refer to one step of this approach as Stochastic Refinement and the general application of multiple iterations as Stochastic Iterative Refinement.

Aligning the model's input and output enables repeated use of the recurrent model. Recurrent models trained on short iterations can be deployed for longer iterations to solve more complex problems (Schwarzschild et al., 2021); however, overuse can result in 'overthinking' (Bansal et al., 2022). $X^2$GNN mitigates this by not iterating over the hidden representations, but rather iterating stochastically over the output space. The stochastic sampling process significantly impacts performance by facilitating exploration around the current solution, allowing exploitation, and escaping local optima. Moreover, it helps generalization to bigger and out-of-distribution datasets.

**Converting Soft Solutions to Hard Solutions:** Since the GNN outputs soft solutions $\hat{Y}^k$, we convert them to discrete feasible solutions $S$ depending on the problem. For MCut, we select a node $u$ into $S$ iff $\hat{Y}^k \geq 0.5$, yielding a feasible solution due to the absence of constraints. For MIS and MC, we add node $u$ into $S$ in order of decreasing probability $\hat{Y}^k_u$ as long as $S \cup \{u\}$ satisfies problem constraints.

**$K$-Coupled Solutions:** To couple solutions, we construct a multilayer graph $G_M = (V', E')$ from the original graph $G = (V, E)$. $G_M$ contains $K$ "layers" each containing a copy of the original graph.

Additional edges (cross edges) connect nodes in different layers corresponding to the same node in $G$. We denote $E'_O$ as the edges corresponding to original edges $E$, and $E'_C$ as the cross edges, such that $E' = E'_O \cup E'_C$. We construct $G_M$ as follows: For each node $v_i \in V$, we create node $v_i^k$ for all layers $k = 1, \ldots, K$. For each edge $(v_i, v_j) \in E$, we create an original edge between $v_i^k$ and $v_j^k$ for all $k = 1, \ldots, K$, forming $E'_O$. We then add cross edges $E'_C$, such that for each node $i \in V$ all copies of $i$ have a pairwise edge between them in $G_M$ In practice, we select $K = 2$.

**$X^2$GNN Neural Network Architecture:** The GNN used by $X^2$GNN to construct and refine solutions consists of $2L$ layers combining Graph Isomorphism Networks (GIN) (Xu et al., 2019) and Graph Attention Networks (GAT) (Velickovic et al., 2018; Brody et al., 2022). The layers alternate between GIN layers operating on $(V', E'_O)$ to work on individual solutions and GAT layers operating on $(V', E'_C)$ to enable information sharing between solutions. This alternating design enables the simultaneous generation of $K$-coupled solutions. Note that the same model parameters are used for both construction and refinement.

**Training and Loss Functions:** We train $X^2$GNN using unsupervised combinatorial optimization losses which take the form of Lagrangian relaxations of the original problem. We adopt nonlinear programming formulations for MCut, MIS, and MC problems from Sanokowski et al. (2024). For MIS and MC, the objective is to include as many nodes in $S$ as possible, penalizing constraint violation, whereas in MCut the objective is to include as many edges in the cut as possible. We additionally propose adding a loss function to promote diversity among the $K$-coupled solutions. In the MIS and MC settings, we want solutions to contain different nodes, whereas we are interested in having different cut edges for MCut. We write the optimization problems, continuous relaxations, and Lagrangian terms for the constraints in Table 1.

Overall, $X^2$GNN trains the GNN parameters $\theta$ to jointly optimize objective quality ($\mathcal{L}_o$), constraint satisfaction ($\mathcal{L}_c$), and solution diversity ($\mathcal{L}_d$) over the training set $\mathcal{G}$:

$$\min_{\theta} \mathbb{E}_{G \in \mathcal{G}} \left[ \sum_t \left[ \sum_k \mathcal{L}_o(G, {}^t\hat{Y}^k) + \lambda_1 \mathcal{L}_c(G, {}^t\hat{Y}^k) \right] + \lambda_2 \mathcal{L}_d(G, {}^t\hat{Y}) \right]$$

For MC and MIS, we impose node diversity:

$$\mathcal{L}_d(G, \hat{Y}) = \frac{1}{K(K-1)} \sum_{\substack{1 \le k_1, k_2 \le K \\ k_1 \ne k_2}} \sum_{u \in V} \hat{Y}_u^{k_1} \hat{Y}_u^{k_2}$$

For MCut, we impose cut edge diversity:

$$\mathcal{L}_d(G, \mathcal{Y}) = \frac{1}{K(K-1)} \sum_{\substack{1 \le k_1, k_2 \le K \\ k_1 \ne k_2}} \sum_{(u,v) \in E} \frac{1 - (2\hat{Y}_u^{k_1} - 1)(2\hat{Y}_v^{k_1} - 1)}{2} \frac{1 - (2\hat{Y}_u^{k_2} - 1)(2\hat{Y}_v^{k_2} - 1)}{2}$$

| Problem | Formulation | | Objective | Constraint loss |
|---------|-------------|---|-----------|-----------------|
| MC | $\max\limits_{Y \in \{0,1\}^{\|V\|}}$ $\sum_{u \in V} Y_u$ 
 s.t. $Y_u Y_v = 0, \ \forall (u,v) \notin E$ | | $\sum_{u \in V} \hat{Y}_u$ | $\sum_{(u,v) \notin E} \hat{Y}_u^k \hat{Y}_v^k$ |
| MIS | $\max\limits_{Y \in \{0,1\}^{\|V\|}}$ $\sum_{u \in V} Y_u$ 
 s.t. $Y_u Y_v = 0, \ \forall (u,v) \in E$ | | $\sum_{u \in V} \hat{Y}_u$ | $\sum_{(u,v) \in E} \hat{Y}_u^k \hat{Y}_v^k$ |
| MCut | $\max\limits_{Y \in \{0,1\}^{\|V\|}} \sum_{(u,v) \in E} \frac{1 - (2Y_u - 1)(2Y_v - 1)}{2}$ | | $\sum_{(u,v) \in E} \frac{1 - (2\hat{Y}_u - 1)(2\hat{Y}_v - 1)}{2}$ | – |

Table 1: Mathematical formulation, objective loss $\mathcal{L}_o$, and constraint loss $\mathcal{L}_c$ for our problems.

We train $X^2$GNN using a two-stage training procedure. In the first stage, the model learns to construct solutions, and in the second stage, the model learns to stochastically refine constructed solutions for one step. The proposed two-stage training procedure leads to better initial solutions and more stable solution refinement for $X^2$GNN.

**Inference:** Unlike during training, during inference we use the stochastic refinement step multiple times leading to stochastic iterative refinement. Training the model on a single stochastic refinement iteration is enough to teach the model to generally improve solutions. We show that using the stochastic iterative refinement longer leads to a significant increase in solution quality. Additionally, instead of generating just one $K$-coupled solution, we generate $C$ $K$-coupled solutions to increase exploration. These $K$-coupled solutions are independent and effectively run $X^2$GNN simultaneously with different random seeds.

$X^2$GNN offers time-quality trade-offs by selecting $C$, the number of $K$-Coupled solutions, and $T$ the number of iterations. By increasing $C$ and/or $T$, we can consider more solutions to improve solution quality by using more time. For each choice of $C$ and $T$, $X^2$GNN generates $C \times K$ solutions at the first iteration that are then refined for $T - 1$ iterations to generate $C \times K \times T$ solutions in total. Hence, the computational impact of $C$ and $T$ is very similar. A natural question is how to select $C$ and $T$ for a fixed computational budget. Under a fixed budget, increasing C promotes exploration by maintaining more solutions, whereas increasing T enhances exploitation by allowing more refinement iterations on existing solutions. This mechanism controls exploration and exploitation in our optimization framework, which is crucial for effectively navigating the search landscape of complex problems.

## 5 EXPERIMENTS

**Datasets:** Previous literature has identified that some problem instances for MIS and MC are relatively easy (Dai et al., 2020). To ensure rigorous evaluation, we use synthetically generated hard instances, following previous work (Karalias and Loukas, 2020; Zhang et al., 2023; Sanokowski et al., 2024). Our datasets include RB graphs (Xu and Li, 2000), a revision to model B graphs (Gent et al., 2001; Smith and Dyer, 1996), known to generate challenging instances for MC and MIS. Specifically, we use RB graphs with 200-300 nodes(RB250) and 800-1200 nodes (RB1000). For MIS, we also evaluate on Erdős-Rényi (ER) graphs (Erdös and Rényi, 1959) with 700-800 nodes and edge probability 0.15 (ER750), as well as regular graphs where each node has either 3 (d=3) or 5 (d=5) neighbors. Following Schuetz et al. (2022), we generate 20 regular graphs for each degree $d \in 3, 5$ and each size n $\in \left[10^2, 10^3, 10^4, 10^5, 10^6\right]$ for testing. To train learning-based methods, we generate 4,000 additional graphs with $n = 10^3$, enabling evaluation of both generalization and scalability. For MCut, we use Barabási-Albert (BA) graphs (Barabási and Albert, 1999) with 250 nodes (BA250) and 1,000 nodes (BA1000). We train on 4,000 graphs and test on 500 graphs except for ER (128 test graphs) and regular graphs (20 test graphs for each size and d).

**Baselines:** We compare $X^2$GNN against both Operations Research (OR) and Machine Learning (ML) techniques. We compare against Gurobi (Gurobi Optimization, LLC, 2024) on all tasks as it is a general-purpose exact solver that is highly performant on many CO tasks due to years of development. For MIS, we compare against KAMIS (Lamm et al., 2016), a highly specialized MIS solver, as well as learning-based approaches such as PPO (Ahn et al., 2020), Gflow (Zhang et al., 2023), DIFFUSCO (Sun and Yang, 2023), T2T (Li et al., 2023), and DiffUCO (Sanokowski et al., 2024). For MC, we benchmark against KAMIS used on the complement graph, greedy algorithms, mean-field annealing (MFA), and learning-based methods including ERDOS and its annealed version ANNEAL (Karalias and Loukas, 2020; Sun et al., 2022), DiffUCO, and Gflow. MCut comparisons include semi-definite-programming (SDP) based approximation algorithm (Goemans and Williamson, 1995), Tabu Search (TS) (Nath and Kuhnle, 2024), and learning-based methods RUN-CSP (Tönshoff et al., 2020), ANYCSP (Tönshoff et al., 2023), ERDOS, ANNEAL, DiffUCO, and Gflow. When given, we use Fast, Quality, and 30min to denote that we set time limits around the twice the fastest version of $X^2$GNN, twice the slowest version of $X^2$GNN, and 30 minutes respectively.

**Evaluation Metrics:** We employ three metrics: the mean objective value (Size), the mean drop in quality relative to the best-known solution (Drop), and the mean runtime in seconds (Time). Overall, better methods find solutions with lower solution quality drop at smaller runtimes. Since all problems are maximization problems, larger size is better. For instance, a 10% drop means the method generates solutions with a mean objective value of 90, while the best method achieves 100. In all tables, learning-based methods are shaded. Bold entries denote the best learning-based method, and italics indicate the best method, learning or traditional. Additionally, we denote the method type categorizing methods into operations research (OR), heuristic (H), supervised learning (SL), and unsupervised learning (UL).

Table 2: Results for Max Clique on small and large RB graphs, presenting the mean clique size, drop in quality compared to the optimal, and runtime in seconds. Learning-based methods are shaded, and the best learning-based result is bolded. The best global result is in italics. $X^2$GNN generates solutions at least 14% to 23% better than all learning-based methods. $X^2$GNN solves RB250 optimally, with a similar run time as Gurobi and KaMIS.

| Method | Type | RB250 | | | RB1000 | | |
|---|---|---|---|---|---|---|---|
| | | Size ↑ | Drop ↓ | Time ↓ | Size ↑ | Drop ↓ | Time ↓ |
| KaMIS | OR | *19.074* | *0%* | 10 | *40.652* | *0%* | 51 |
| Gurobi (30min) | OR | *19.074* | *0%* | 0.73 | *40.652* | *0%* | 287 |
| Gurobi (Quality) | OR | 19.068 | 0.03% | 0.61 | 36.23 | 10.88% | 47 |
| Gurobi (Fast) | OR | 14.62 | 23.35% | 0.17 | 25.36 | 37.62% | 3 |
| Greedy | H | 13.53 | 29.07% | 0.03 | 26.71 | 34.30% | 0.04 |
| MFA | H | 14.82 | 22.30% | 0.04 | 27.94 | 31.27% | 0.21 |
| Erdos | UL | 12.02 | 36.98% | 0.06 | 25.43 | 37.44% | 0.2 |
| Anneal | UL | 14.1 | 26.08% | 0.06 | 27.46 | 32.45% | 0.2 |
| Gflow | UL | 16.24 | 14.86% | 0.06 | 31.42 | 22.71% | 0.44 |
| DiffUCO | UL | 16.3 | 14.54% | 4.13 | 30.5 | 24.97% | 7.92 |
| $X^2$GNN (RB250)(2x64) | UL | 19.04 | 0.18% | 0.09 | 39.83 | 2.02% | 1.5 |
| $X^2$GNN (RB250)(8x64) | UL | 19.072 | 0.01% | 0.37 | 40.09 | 1.38% | 5.8 |
| $X^2$GNN (RB250)(32x64) | UL | ***19.074*** | ***0%*** | 1.41 | **40.17** | **1.19%** | 23.5 |

For generalization, we denote the training dataset with (RB250) or (BA250) in the model name. We denote variants of $X^2$GNN that generate $C$ 2-Coupled solutions and use $T - 1$ stochastic refinement steps with (2CxT) in the model name.

## 5.1 RESULTS ON MAXIMUM CLIQUE

Results for MC on small and large RB datasets are shown in Table 2. For MC, we train $X^2$GNN on RB250 and showcase generalization to larger instances. $X^2$GNN generates solutions of at least 14% and 23% higher objective value than the second-best learning-based methods on RB250 and RB1000, respectively.

Compared to traditional algorithms, $X^2$GNN solves every RB250 instance optimally, with a similar run time as Gurobi and KaMIS. On the larger dataset, on which $X^2$GNN wasn't trained, $X^2$GNN generates solutions that are within 2% of optimality while almost being 50 and 190 times faster than KaMIS and Gurobi, respectively. Additionally, at a similar runtime, $X^2$GNN has substantially better solution quality than Gurobi.

## 5.2 RESULTS ON MAXIMUM INDEPENDENT SET

Table 3 presents results for Maximum Independent Set (MIS) on small RB, large RB, and ER graphs. For all datasets except regular graphs, the metaheuristic KaMIS achieves the best solution quality. Again, $X^2$GNN outperforms all learning-based methods by a large margin, especially for the largest RB1000 dataset, where $X^2$GNN generates solutions that are 9% better than the second best. Even the model trained on RB250 dataset is able to outperform the other learning-based methods on both RB1000 and ER750 datasets, showing that $X^2$GNN can successfully generalize to harder and different graph distributions.

Comparison with the traditional algorithms is more nuanced. On ER750, $X^2$GNN generates better solutions than KaMIS and Gurobi when they are given either 15 or 60 seconds. On RB1000, $X^2$GNN generates better solutions than Gurobi but slightly worse solutions than KaMIS at around 20 seconds.

Figure 2 presents the solution quality relative to the theoretical lower bound and the running time for regular graphs of varying sizes. Given the large graph sizes, obtaining optimal solutions is intractable. Following Schuetz et al. (2022), we use analytical upper bounds for random d-regular graphs, with the best-known ratios $\alpha_3/n = 0.45537$ and $\alpha_5/n = 0.38443$ for $d = 3$ and $d = 5$, respectively (Duckworth and Zito, 2009). However, these bounds may not be tight. For instances with $n = 100$ and $d = 3$, both $X^2$GNN and KaMIS find optimal solutions, as verified by Gurobi. Notably, the analytical upper bound remains 2.6% above these solutions, indicating that actual performance may be better than suggested.

Table 3: Results for Max Independent Set on small and large RB graphs and ER graphs, presenting the mean independent set size, drop in quality compared to the virtual best (KaMIS), and runtime in seconds. $X^2$GNN substantially outperforms learning-based approaches on all datasets. When generalizing from small RB250 instances, $X^2$GNN outperforms learning-based methods trained on the larger and in-distribution problems. On ER instances, $X^2$GNN outperforms traditional OR approaches given similar time limits.

| Method | Type | RB250 | | | RB1000 | | | ER750 | | |
|---|---|---|---|---|---|---|---|---|---|---|
| | | Size ↑ | Drop ↓ | Time ↓ | Size ↑ | Drop ↓ | Time ↓ | Size ↑ | Drop ↓ | Time ↓ |
| KaMIS (30min) | OR | *20.106* | *0%* | 3.92 | *43.218* | *0%* | 381 | *45.234* | *0%* | 382 |
| Gurobi (30min) | OR | *20.106* | *0%* | 0.31 | 42.96 | 0.60% | 550 | 43.62 | 3.57% | 1800 |
| KaMIS (Quality) | OR | *20.106* | *0%* | 3.92 | 42.98 | 0.55% | 18 | 44.84 | 0.87% | 61 |
| Gurobi (Quality) | OR | *20.106* | *0%* | 0.42 | 42.25 | 2.24% | 22.27 | 43.5 | 3.83% | 120 |
| KaMIS (Fast) | OR | 20.032 | 0.37% | 1.16 | 42.66 | 1.29% | 6.5 | 43.46 | 3.92% | 16 |
| Gurobi (Fast) | OR | 19.16 | 4.71% | 0.1 | 38.81 | 10.20% | 1.23 | 41.31 | 8.67% | 3.62 |
| PPO | UL | 19.01 | 5.45% | 0.15 | 32.32 | 25.22% | 0.91 | 41.11 | 9.12% | 2.11 |
| GFlow | UL | 19.18 | 4.61% | 0.05 | 37.48 | 13.28% | 0.4 | 41.14 | 9.05% | 1.03 |
| DIFUSCO | SL | 17.68 | 12.07% | 0.87 | 35.82 | 17.12% | 41.11 | 40.35 | 10.80% | 15.46 |
| T2T | SL | 18.35 | 8.73% | 2.32 | 35.822 | 17.11% | 26.55 | 41.37 | 8.54% | 13.92 |
| DiffUCO | UL | 19.24 | 4.31% | 0.42 | 38.87 | 10.06% | 5 | 43.63 | 3.55% | 0.71 |
| $X^2$GNN(16x8) | UL | 19.51 | 2.96% | 0.034 | 40.53 | 6.22% | 0.3 | 42.05 | 7.04% | 0.31 |
| $X^2$GNN(64x8) | UL | 19.82 | 1.42% | 0.128 | 41.54 | 3.88% | 1.18 | 43.06 | 4.81% | 1.07 |
| $X^2$GNN(256x8) | UL | 19.98 | 0.63% | 0.5 | 42.19 | 2.38% | 4.66 | 43.82 | 3.13% | 3.91 |
| $X^2$GNN(256x32) | UL | 20.072 | 0.17% | 1.94 | 42.48 | 1.71% | 18.36 | 44.43 | 1.78% | 15.26 |
| $X^2$GNN(1024x32) | UL | **20.098** | **0.04%** | 7.11 | **42.81** | **0.94%** | 74.4 | **44.91** | **0.72%** | 57.18 |
| $X^2$GNN(RB250)(256x32) | UL | 20.072 | 0.17% | 1.94 | 39.28 | 9.1% | 9.43 | 44.15 | 2.40% | 7.96 |

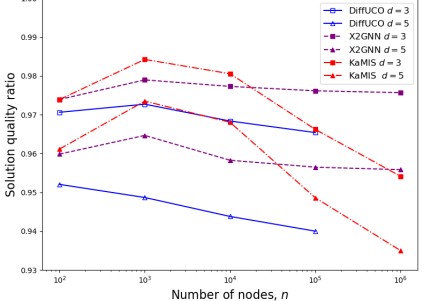 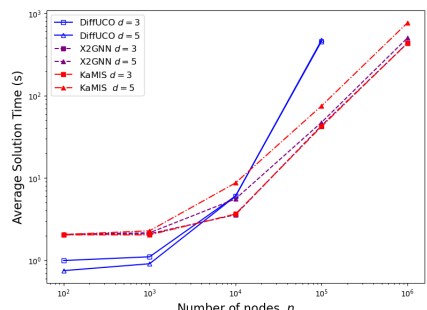

Figure 2: The left plot shows the relative solution quality compared to the theoretical upper bound for $X^2$GNN, DiffUCO, and KaMIS (higher is better). The right plot shows the average running time for each method (lower is better). Lines with squares and triangles show the results for regular graphs where each node has 3 and 5 neighbors, respectively. $X^2$GNN outperforms DiffUCO on all graph sizes and families in terms of solution quality while being faster on larger graphs. It also surpasses KaMIS in larger graphs with a better run time.

$X^2$GNN surpasses DiffUCO in both solution quality and computational efficiency on larger graphs. DiffUCO fails to solve $n = 10^6$ graphs within two hours, while $X^2$GNN efficiently produces high-quality solutions in about 600 seconds. Although KaMIS slightly outperforms $X^2$GNN on smaller instances, $X^2$GNN excels on larger graphs with lower computational time, scaling effectively to instances three orders of magnitude larger than those in training.

## 5.3 RESULTS ON MAXIMUM CUT

Results for Maximum Cut on small and large BA datasets are shown in Table 4. $X^2$GNN outperforms all state-of-the-art learning-based methods on both datasets. However, ANYCSP and DiffUCO notably have only slightly worse performance than $X^2$GNN both in terms of speed and quality. For the large dataset, $X^2$GNN outperforms Gurobi with a time limit of 30 minutes per instance while only using 0.2 seconds. Similarly, $X^2$GNN outperforms Tabu Search, finding better solutions faster.

Table 4: Results for Max Cut on small and large BA graphs, presenting the mean cut size, drop in quality to the virtual best, and runtime in seconds. $X^2$GNN outperforms learning-based methods, slightly outperforming ANYCSP. Additionally, on BA1000, $X^2$GNN outperforms Gurobi given 30 minutes by generating better solutions in 0.2s.

| Method | Type | BA250 | | | BA1000 | | |
|---|---|---|---|---|---|---|---|
| | | Size ↑ | Drop ↓ | Time ↓ | Size ↑ | Drop ↓ | Time ↓ |
| Gurobi (30min) | OR | *735.32* | *0%* | 759 | 2966.58 | 0.76% | 1800 |
| Gurobi (Quality) | OR | 731.99 | 0.45% | 2.5 | 2931.02 | 1.95% | 4.24 |
| Gurobi (Fast) | OR | 731.83 | 0.47% | 0.16 | 2930.99 | 1.95% | 0.66 |
| SDP | OR | 700.04 | 4.8% | 4.2 | - | - | - |
| Tabu Search | H | 733.79 | 0.21% | 3 | 2926.6 | 2.10% | 11.5 |
| Greedy | H | 688.31 | 6.39% | 0.02 | 2761.06 | 7.64% | 0.29 |
| MFA | H | 704.03 | 4.26% | 0.15 | 2833.86 | 5.20% | 0.66 |
| Erdos | UL | 693.45 | 5.69% | 0.07 | 2870.34 | 3.98% | 0.25 |
| Anneal | UL | 696.73 | 5.25% | 0.07 | 2863.23 | 4.22% | 0.24 |
| GFlow | UL | 704.3 | 4.22% | 0.27 | 2864.61 | 4.17% | 1.95 |
| RUN-CSP | UL | 726.96 | 1.14% | 0.78 | 2925.8 | 2.12% | 10.8 |
| ANYCSP | UL | 735.12 | 0.03% | 2.5 | 2988.6 | 0.02% | 7.37 |
| DiffUCO | UL | 733.5 | 0.25% | 2.77 | 2981.1 | 0.27% | 6.6 |
| $X^2$GNN(16x8) | UL | 734.21 | 0.15% | 0.08 | 2985.2 | 0.14% | 0.2 |
| $X^2$GNN(64x8) | UL | 734.92 | 0.05% | 0.15 | 2987.7 | 0.05% | 0.48 |
| $X^2$GNN(256x8) | UL | 735.17 | 0.02% | 1.2 | 2988.7 | 0.02% | 1.95 |
| $X^2$GNN(256x32) | UL | **735.26** | **0.01%** | 4.1 | ***2989.3*** | *0%* | 7.3 |
| $X^2$GNN (BA250)(256x32) | UL | **735.26** | **0.01%** | 4.1 | 2985.5 | 0.13% | 5.4 |

These results indicate that on 3 CO problems, $X^2$GNN outperforms neural baselines, is competitive with specialized metaheuristics like KaMIS, and improves over general solvers like Gurobi.

## 5.4 ABLATION

In this section, we analyze the impact of $K$-coupled solutions for different values of $K$. We also measure the impact of stochastic refinement, two-stage training, and encouraging diversity.

Table 5: Effects of the parameter $K$ on $K$-coupled solutions, showing $K = 2$ gives the lowest drop.

| Problem | $K = 1$ | $K = 2$ | $K = 4$ | $K = 8$ |
|---|---|---|---|---|
| MIS | 4.01% | 0.43% | 3.81% | 9.47% |
| MC | 0.14% | 0.01% | 0.21% | 1.79% |
| MCut | 0.030% | 0.015% | 0.021% | 0.157 % |

Table 5 demonstrates that $K = 2$ is the optimal choice for each problem. Its impact is particularly significant for MIS, while more subtle for MC and MCut.

We evaluate the impact of ablating aspects of $X^2$GNN by comparing to our standard version which achieves a drop value of 0.43% on MIS. Using deterministic refinement instead of stochastic refinement significantly increases the drop value to 4.97%. Training the full framework in a single stage increased the drop value to 1.44%. Similarly, ignoring the diversity loss raises the drop value to 1.81%. These findings highlight the cumulative benefits of our proposed techniques. The combination of $K = 2$ coupled solutions, stochastic refinement, two-stage training, and diversity loss is crucial for the superior performance of $X^2$GNN.

## 5.5 NEURAL SEARCH DYNAMICS

For a fixed budget, $X^2$GNN can controllably balance exploration and exploitation by trading off the number of solution couples generated at each iteration $C$, with the number of iterations $T$ taken.

MC and MIS problems benefit from different exploration-exploitation trade-offs due to their distinct feasible region structures: In MC, selecting a node restricts subsequent additions to only its neighbors, causing the search space to contract rapidly with each decision. Early choices have high impact, potentially eliminating optimal solutions if poor selections are made. Consequently, exploring diverse starting points (higher $C$) becomes crucial. In MIS, selecting a node only eliminates its neighbors

from consideration, typically leaving a substantial portion of nodes available. This more gradual search space reduction allows solutions to recover from suboptimal early choices through additional refinement iterations.

As shown in Figure 3, our experiments confirm these insights. Figure 3a demonstrates that MC benefits from prioritizing exploration (higher C), with C=64 providing optimal performance. Conversely, Figure 3b shows that MIS performs better with emphasis on exploitation (higher T, lower C), with C=4 yielding the best results.

These findings highlight that while all combinatorial optimization problems require both strategies, their relative importance varies based on how solution construction progressively constrains the search space.

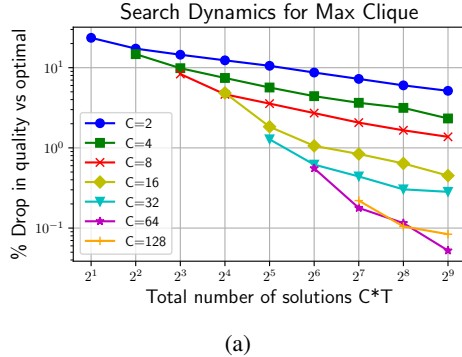 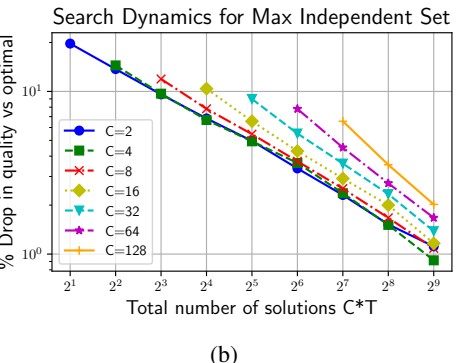

(a)                                             (b)

Figure 3: Search dynamics on RB250: Each line corresponds to one setting of $C$, where higher values increase exploration. We present the drop in solution quality (lower is better) over various computational budgets. For MC, higher values of $C$ yield better performance, suggesting the search benefits from broader exploration of the solution space. Conversely, for MIS, lower values of $C$ lead to better solutions, indicating the search benefits from deeper exploitation of promising regions.

## 6 Conclusion

In this work, we introduce Explore-and-Exploit GNN ($X^2$GNN), a novel unsupervised neural framework that addresses a key challenge in learning-based combinatorial optimization (CO). Unlike most existing approaches that focus on constructing a limited number of solutions, $X^2$GNN effectively explores the vast search space of NP-hard CO problems through two key mechanisms:

(i) Exploration: $X^2$GNN generates multiple solutions simultaneously, and promotes solution diversity.

(ii) Exploitation: $X^2$GNN employs neural stochastic iterative refinement, using sampled partial solutions to guide the search toward promising regions and escape local optima.

Our experiments on three canonical NP-Hard CO problems - Maximum Clique (MC), Maximum Independent Set (MIS), and Maximum Cut (MCut) - demonstrate that $X^2$GNN significantly outperforms state-of-the-art learning-based approaches. Notably, for large MC problems, $X^2$GNN consistently generates solutions within 1.2% of optimality, while other learning-based methods struggle to reach within 22% of optimal. Moreover, $X^2$GNN exhibits exceptional generalization capabilities, outperforming existing methods even when trained on smaller or out-of-distribution graphs. The iterative nature of $X^2$GNN allows users to trade off runtime and solution quality, as the model can be applied indefinitely to refine solutions. This feature, combined with its strong performance and generalization performance, positions $X^2$GNN as a competitive framework with promising directions for future research in learning-based heuristics for combinatorial optimization. By balancing exploration and exploitation, $X^2$GNN offers a more effective and adaptable approach to neural combinatorial optimization, addressing the limitations of existing methods and paving the way for more robust solutions to complex CO problems across various domains.

ACKNOWLEDGMENTS

This project is partially supported by Schmidt Sciences programs, an AI2050 Senior Fellowship and two Eric and Wendy Schmidt AI in Science Postdoctoral Fellowships; the National Science Foundation (NSF); the National Institute of Food and Agriculture (USDA/NIFA); the Air Force Office of Scientific Research (AFOSR). Additionally, Utku Umur Acikalin is supported by the Turkish Ministry of National Education.

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

## A EXPERIMENTAL DETAILS

### A.1 NETWORK ARCHITECTURE

We use a GNN architecture where Graph Isomorphism Network (GIN) (Xu et al. (2019)) and Graph Attention Network (GAT) (Velickovic et al. (2018); Brody et al. (2022)) layers are used in an interleaved manner. We use $n_L$ GIN and GAT layers. The initial node features (node probabilities) are transformed to a higher dimension by a linear layer before they are passed to the first GIN layer.

The GIN layers work on the original edges and GAT layers work on the crossedges. GIN layers are composed of two-layer MLPs with RELU activations after the MLP layers. We apply Layer Normalization after each MLP layer in GINs and after each GAT layer. We add skip connections to both GIN and GAT layers. Finally, the output of each GAT layer is concatenated and fed into a final two-layer MLP. This architecture is executed recurrently starting from the last hidden representation from MLP $n_R$ times. The final output is transformed into logits and the probability that each node belongs to S is computed via Softmax. For all problems, the architecture hidden representation size is 64.

### A.2 HYPERPARAMETERS

All the hyperparameters on each dataset is given in Table 6. $n_L$ and $n_R$ represents the number of layers and number of recurrent steps. $K$ and $C$ means $C$ $K-$Coupled solutions are generated during training. On all settings, we select $K = 2$ and $C = 4$ during training. $K$ and $C$ values used during inference are shown right next to the model. $\lambda_1$ and $\lambda_2$ are the weights of the constraint and diversity loss, respectively.

| Dataset | ep | lr | bs | $n_L$ | $n_R$ | h | K | C | $\phi$ | $\lambda_1$ | $\lambda_2$ |
|---|---|---|---|---|---|---|---|---|---|---|---|
| RB-250 MIS | 50 | 0.001 | 64 | 4 | 2 | 64 | 2 | 4 | 0.5 | 1 | 0.75 |
| RB-1000 MIS | 50 | 0.001 | 64 | 4 | 4 | 64 | 2 | 4 | 0.5 | 1 | 0.75 |
| ER-750 MIS | 50 | 0.001 | 64 | 4 | 4 | 64 | 2 | 4 | 0.5 | 1 | 0.75 |
| RB-250 MC | 50 | 0.001 | 64 | 4 | 2 | 64 | 2 | 4 | 0.8 | 1 | 0.75 |
| BA-250 MCut | 50 | 0.001 | 64 | 4 | 2 | 64 | 2 | 4 | 0.8 | - | 0.75 |
| BA-1000 MCut | 50 | 0.001 | 64 | 4 | 4 | 64 | 2 | 4 | 0.8 | - | 0.75 |

Table 6: Hyperparameters used in training.

### A.3 COMPUTATIONAL RESOURCES AND TIME MEASUREMENTS

All experiments are conducted on a machine with 2 Intel Xeon 6348 processors with 28 cores, 1TB of memory, and 4 A100 80GB GPUs. All traditional heuristics are run using a single core and Gurobi is run with 4 cores, allowing 40GB of memory for each instance.

We only use 1 A100 during training or inference for all datasets. The training time for MIS is roughly 1 hour and 9 hours for RB250 and RB1000 datasets, respectively. The training time for MCut is roughly 1 hour and 3 hours for BA250 and BA1000 datasets, respectively. The training time for MC is roughly 2 hours for the RB250 dataset.

For baselines, we run each method in our machine and report the time we obtain on our machine to have a fair comparison. However, we use the original size when available on the same dataset split.

## B ADDITIONAL EXPERIMENTS

### B.1 ADDITIONAL EXPERIMENTS FOR MAXIMUM CLIQUE

For MC, we consider a challenging dataset from DIMACS implementation challenges related Maximum Clique (DIMACS). This dataset contains 37 challenging graphs. We use the $X^2$GNN model trained on the RB-1000 dataset as well as its fine-tuned version. We report the performance of $X^2$GNN 256x32 with and without fine-tuning as well as the best traditional heuristic KaMIS.

Table 7: Table shows the clique size and the running time for each instance for $X^2$GNN and KaMIS, considering that $X^2$GNN FT fine-tunes $X^2$GNN on these instances, and $X^2$GNN is trained on only RB1000 instances. The clique sizes matching the best-known solutions are shown in bold.

| Instance | n | m | Best known | Size KaMIS | $X^2$GNN FT | $X^2$GNN | Runnig Time KaMIS | $X^2$GNN FT | $X^2$GNN |
|---|---|---|---|---|---|---|---|---|---|
| brock200_2 | 200 | 9,876 | 12 | **12** | 11 | 11 | 7.14 | 4.95 | 4.95 |
| brock200_4 | 200 | 13,089 | 17 | 16 | 16 | 16 | 5.45 | 3.7 | 3.67 |
| brock400_2 | 400 | 59,786 | 29 | 24 | 24 | 24 | 9.17 | 4.32 | 4.35 |
| brock400_4 | 400 | 59,765 | 33 | 25 | 25 | 25 | 8.77 | 4.28 | 4.29 |
| brock800_2 | 800 | 208,166 | 24 | 20 | 20 | 20 | 33.91 | 10.84 | 10.79 |
| brock800_4 | 800 | 207,643 | 26 | 20 | 21 | 21 | 36.02 | 10.87 | 10.8 |
| C1000.9 | 1,000 | 450,079 | 68 | 66 | 66 | 64 | 19.06 | 6.68 | 6.59 |
| C125.9 | 125 | 6,963 | 34 | **34** | **34** | **34** | 5.26 | 3.72 | 3.6 |
| C2000.5 | 2,000 | 999,836 | 16 | 15 | 14 | 0 | 188.80 | 72.63 | 72.4 |
| C2000.9 | 2,000 | 1,799,532 | 80 | 75 | 75 | 73 | 45.28 | 17.67 | 17.59 |
| C250.9 | 250 | 27,984 | 44 | **44** | **44** | **44** | 5.15 | 3.72 | 3.65 |
| C4000.5 | 4,000 | 4,000,268 | 18 | 16 | 13 | 0 | 596.32 | 367.1 | 369.75 |
| C500.9 | 500 | 112,332 | 57 | 56 | 56 | 55 | 6.88 | 3.8 | 3.74 |
| DSJC1000_5 | 1,000 | 499,652 | 15 | **15** | 12 | 12 | 70.18 | 20.1 | 19.96 |
| DSJC500_5 | 500 | 125,248 | 13 | **13** | **13** | **13** | 26.61 | 7.29 | 7.24 |
| gen200_p0.9_44 | 200 | 17,910 | 44 | **44** | **44** | **44** | 5.20 | 3.71 | 3.59 |
| gen200_p0.9_55 | 200 | 17,910 | 55 | **55** | **55** | **55** | 5.20 | 3.74 | 3.6 |
| gen400_p0.9_55 | 400 | 71,820 | 55 | 53 | **55** | **55** | 5.35 | 3.71 | 3.58 |
| gen400_p0.9_65 | 400 | 71,820 | 65 | **65** | **65** | **65** | 5.22 | 3.71 | 3.58 |
| gen400_p0.9_75 | 400 | 71,820 | 75 | **75** | **75** | **75** | 5.65 | 3.75 | 3.58 |
| hamming10-4 | 1,024 | 434,176 | 40 | 38 | **40** | 38 | 12.24 | 9.53 | 9.4 |
| hamming8-4 | 256 | 20,864 | 16 | **16** | **16** | **16** | 5.76 | 3.66 | 3.55 |
| keller4 | 171 | 9,435 | 11 | **11** | **11** | **11** | 5.40 | 3.71 | 3.58 |
| keller5 | 776 | 225,990 | 27 | 26 | 23 | **27** | 10.97 | 8.32 | 8.26 |
| keller6 | 3,361 | 4,619,898 | 59 | 55 | 39 | 43 | 61.87 | 75.01 | 75.2 |
| MANN_a27 | 378 | 70,551 | 126 | **126** | **126** | 92 | 5.05 | 3.69 | 3.57 |
| MANN_a45 | 1,035 | 533,115 | 345 | 344 | 343 | 244 | 5.09 | 3.7 | 3.59 |
| MANN_a81 | 3,321 | 5,506,380 | 1100 | **1100** | 1097 | 664 | 5.06 | 4.25 | 4.14 |
| p_hat1500-1 | 1,500 | 284,923 | 12 | 11 | 10 | 0 | 88.97 | 61.61 | 60.96 |
| p_hat1500-2 | 1,500 | 568,960 | 65 | **65** | **65** | 62 | 85.44 | 39.96 | 40.47 |
| p_hat1500-3 | 1,500 | 847,244 | 94 | **94** | **94** | **94** | 44.08 | 22.49 | 22.45 |
| p_hat300-1 | 300 | 10,933 | 8 | **8** | **8** | **8** | 18.69 | 5.27 | 5.19 |
| p_hat300-2 | 300 | 21,928 | 25 | **25** | **25** | **25** | 12.02 | 4.48 | 4.38 |
| p_hat300-3 | 300 | 33,390 | 36 | **36** | **36** | **36** | 7.06 | 3.75 | 3.57 |
| p_hat700-1 | 700 | 60,999 | 11 | **11** | 9 | 2 | 42.16 | 15.36 | 15.27 |
| p_hat700-2 | 700 | 121,728 | 44 | **44** | **44** | **44** | 25.06 | 11.49 | 11.42 |
| p_hat700-3 | 700 | 183,010 | 62 | **62** | **62** | **62** | 13.62 | 7.38 | 7.31 |
| | | | | 3.87% | 6.68% | 17.41% | 15.60 | 8.46 | 8.33 |

Table 7 shows that $X^2$GNN is able to generalize and generate optimal or near-optimal solutions on many instances even though it is trained on RB1000 instances, a set of much sparser instances. However, it does fail to generate good solutions on a few instances. With fine-tuning, solution quality improves quite a lot both for these failing cases and also in general, leading to an average gap of 6.68% from 17.41%. Even the most successful traditional heuristic KaMIS achieves an average gap of 3.87%. Considering this dataset is designed to be a challenging dataset, learning a general rule based on RB1000 instances that can solve many problems, and achieving a gap of 6.68% after fine-tuning is noteworthy.

## B.2 ADDITIONAL EXPERIMENTS FOR MAXIMUM INDEPENDENT SET

For MIS, we evaluate $X^2$GNN's performance on an additional dataset that comprises instances from Coding Theory applications, specifically error correction codes, with graph sizes ranging from 64 to 4,096 nodes. For this dataset, we utilize the $X^2$GNN model trained on the RB1000 dataset. We compare against KaMIS, the state-of-the-art MIS solver.

On the Coding Theory dataset (Table 8), $X^2$GNN finds the best-known solution in 20 of 32 instances, compared to KaMIS's 28 instances. $X^2$GNN achieves an average gap of 3.37% from best-known solutions (versus 0.34% for KaMIS). Excluding one outlier instance where $X^2$GNN finds a solution less than half the best-known value, $X^2$GNN's average gap improves to 1.74%. This performance is notable given no domain-specific tuning was performed.

To demonstrate $X^2$GNN's adaptability to weighted problems, we extend it to weighted maximum independent set problems with minimal modifications: adding a weight embedding layer combined

Table 8: The table shows the size of the independent sets found by $X^2$GNN and KaMIS and the running time in seconds for each instance. The last row shows the average gap in percentages from the best-known solution. The instances where a method found the best-known solutions are shown in bold.

| Graph | Best Known | $X^2$GNN 256x32 | KaMIS | $X^2$GNN 256x32 Time | KaMIS Time |
|---|---|---|---|---|---|
| 1dc.64 | 10 | **10** | **10** | 5.06 | 5.14 |
| 1dc.128 | 16 | **16** | **16** | 3.87 | 4.04 |
| 1dc.256 | 30 | **30** | **30** | 4.37 | 4.50 |
| 1dc.512 | 52 | **52** | **52** | 6.33 | 6.49 |
| 1dc.1024 | 94 | **94** | 93 | 11.29 | 11.40 |
| 1dc.2048 | 172 | **172** | **172** | 24.25 | 24.40 |
| 1et.64 | 18 | **18** | **18** | 3.81 | 1.31 |
| 1et.128 | 28 | **28** | **28** | 3.81 | 3.85 |
| 1et.256 | 50 | **50** | **50** | 3.82 | 3.90 |
| 1et.512 | 100 | 98 | **100** | 4.68 | 4.74 |
| 1et.1024 | 171 | 165 | **171** | 7.12 | 7.23 |
| 1et.2048 | 316 | 300 | **316** | 13.95 | 14.02 |
| 1tc.8 | 4 | **4** | **4** | 3.71 | 0.48 |
| 1tc.16 | 8 | **8** | **8** | 3.76 | 0.81 |
| 1tc.32 | 12 | **12** | **12** | 3.76 | 0.30 |
| 1tc.64 | 20 | **20** | **20** | 3.79 | 1.10 |
| 1tc.128 | 38 | **38** | **38** | 3.75 | 1.15 |
| 1tc.256 | 64 | 63 | 63 | 3.78 | 3.88 |
| 1tc.512 | 110 | 109 | **110** | 4.39 | 4.48 |
| 1tc.1024 | 196 | 189 | **196** | 6.64 | 6.73 |
| 1tc.2048 | 352 | 332 | **352** | 12.91 | 13.00 |
| 1zc.128 | 18 | **18** | **18** | 3.77 | 3.87 |
| 1zc.256 | 36 | **36** | **36** | 3.99 | 4.10 |
| 1zc.512 | 62 | **62** | **62** | 5.46 | 5.55 |
| 1zc.1024 | 112 | 109 | **112** | 9.13 | 9.32 |
| 1zc.2048 | 198 | 181 | 195 | 18.78 | 18.89 |
| 1zc.4096 | 379 | 326 | 353 | 40.19 | 40.41 |
| 2dc.128 | 5 | **5** | **5** | 4.52 | 1.46 |
| 2dc.256 | 7 | **7** | **7** | 8.22 | 10.06 |
| 2dc.512 | 11 | **11** | **11** | 19.07 | 22.31 |
| 2dc.1024 | 16 | 15 | **16** | 52.72 | 57.81 |
| 2dc.2048 | 24 | 11 | **24** | 152.61 | 162.73 |
| Average Gap | | 3.37% | 0.34% | | |

with node representations via summation, and incorporating weights into the loss function calculation. Using the RB250 dataset with uniform random integer weights between 1 and 5, we compare against optimal solutions from Gurobi. Table 9 shows $X^2$GNN maintains high solution quality with a 0.8% optimality gap.

Table 9: Results for Weighted Maximum Independent Set on small RB graphs, presenting the mean independent set size, drop in quality compared to the optimal, and run time in seconds.

| Method | Type | RB250 | | |
|---|---|---|---|---|
| | | Size ↑ | Drop ↓ | Time ↓ |
| Gurobi | OR | *82.94* | *0%* | 0.21 |
| $X^2$GNN(16x8) | UL | 78.52 | 5.33% | 0.03 |
| $X^2$GNN(64x8) | UL | 80.26 | 3.23% | 0.09 |
| $X^2$GNN(256x8) | UL | 81.31 | 1.97% | 0.35 |
| $X^2$GNN(256x32) | UL | 81.83 | 1.34% | 1.25 |
| $X^2$GNN(1024x32) | UL | **82.28** | **0.8%** | 4.92 |

In an ablation study, we replaced the GAT layer with a simple MLP for processing cross-edges. Table 10 shows this variant still outperforms other learning-based approaches but underperforms compared to the GAT version, indicating GAT's superior capability in aggregating information across different solutions.

Table 10: The results for replacing GAT layers with a simple MLP layer.

| Method | Type | RB250 | | |
|---|---|---|---|---|
| | | Size ↑ | Drop ↓ | Time ↓ |
| KaMIS (30min) | OR | *20.106* | *0%* | 3.92 |
| Gurobi (30min) | OR | *20.106* | *0%* | 0.31 |
| KaMIS (Quality) | OR | *20.106* | *0%* | 3.92 |
| Gurobi (Quality) | OR | *20.106* | *0%* | 0.42 |
| KaMIS (Fast) | OR | 20.032 | 0.37% | 1.16 |
| Gurobi (Fast) | OR | 19.16 | 4.71% | 0.1 |
| PPO | UL | 19.01 | 5.45% | 0.15 |
| GFlow | UL | 19.18 | 4.61% | 0.05 |
| DIFUSCO | SL | 17.68 | 12.07% | 0.87 |
| T2T | SL | 18.35 | 8.73% | 2.32 |
| DiffUCO | UL | 19.24 | 4.31% | 0.42 |
| $X^2$GNN-GAT(16x8) | UL | 19.51 | 2.96% | 0.034 |
| $X^2$GNN-GAT(64x8) | UL | 19.82 | 1.42% | 0.128 |
| $X^2$GNN-GAT(256x8) | UL | 19.98 | 0.63% | 0.5 |
| $X^2$GNN-GAT(256x32) | UL | 20.072 | 0.17% | 1.94 |
| $X^2$GNN-GAT(1024x32) | UL | **20.098** | **0.04%** | 7.11 |
| $X^2$GNN-MLP(16x8) | UL | 18.95 | 5.75% | 0.027 |
| $X^2$GNN-MLP(64x8) | UL | 19.41 | 3.46% | 0.096 |
| $X^2$GNN-MLP(256x8) | UL | 19.698 | 2.03% | 0.38 |
| $X^2$GNN-MLP(256x32) | UL | 19.886 | 1.09% | 1.35 |

## B.3 ADDITIONAL EXPERIMENTS FOR MCUT

For the Maximum Cut (MCut) problem, we evaluate $X^2$GNN's out-of-distribution performance on two additional benchmark sets. The first dataset, introduced in Mirka and Williamson (2023), comprises diverse graphs from the SNAP Networks repository (referred to as SNAP dataset). This dataset is particularly suitable for assessing generalization capabilities due to its heterogeneous graph distributions. The second dataset, known as Gset [2], is a well-established benchmark collection traditionally used to evaluate MCut algorithms.

To train $X^2$GNN, we generate 4,000 Erdős-Rényi (ER) graphs with sizes uniformly sampled from between 200 and 500 and edge probabilities from [0.1, 0.75]. We then evaluate the trained model on both SNAP and Gset datasets.

Table 11 presents results for the SNAP dataset, comparing $X^2$GNN against ANYCSP (trained on the same dataset), Tabu Search (TS), Semidefinite Programming (SDP) relaxation, and BMZ heuristic from Mirka and Williamson (2023). $X^2$GNN discovers the best solutions among all compared methods for all but two instances, demonstrating superior performance over both ANYCSP and traditional heuristics.

For the Gset evaluation, we use on all unweighted GSET instances, demonstrating that $X^2$GNN can generalize to larger instances. Table 12 shows the cut size achieved by the traditional heuristics BLS, DSDP, KHLWG and learning based heuristics X2GNN and ANYCSP for each instance. To aggregate the results, we first compute the virtual best solution by taking the maximum value found by any of the compared algorithms. Then, for each instance we compute the average gap from the best as 1 - solution/best and take the mean to calculate the average gap from the virtual best solution.

---

[2]https://web.stanford.edu/~yyye/yyye/Gset/

Table 11: Table shows the cut sizes for instance and method. The cut sizes matching the best are shown in bold.

| Graph | n | m | $X^2$GNN (256x32) | ANYCSP | TS | BMZ | SDP |
|---|---|---|---|---|---|---|---|
| ENZYMES8 | 88 | 133 | **126** | **126** | **126** | **126** | **126** |
| johnson16-2-4 | 120 | 5460 | **3036** | 2941 | **3036** | **3036** | **3036** |
| hamming6-2 | 64 | 1824 | **992** | 946 | **992** | **992** | **992** |
| ia-infect-hyper | 113 | 2196 | **1279** | 1208 | **1279** | 1278 | 1275 |
| soc-dolphins | 62 | 159 | **122** | **122** | **122** | **122** | **122** |
| email-enron-only | 143 | 623 | **427** | **427** | **427** | 426 | 422 |
| dwt_209 | 209 | 976 | **557** | **557** | **557** | 557 | 551 |
| ca-netscience | 379 | 914 | 620 | 580 | 627 | 634 | 634 |
| ia-infect-dublin | 410 | 2765 | **1771** | 1709 | 1758 | 1767 | 1750 |
| road-chesapeake | 39 | 170 | **126** | 126 | **126** | 126 | 125 |
| Erdos991 | 492 | 1417 | **1036** | **1036** | 1012 | 1031 | 1019 |
| dwt_503 | 503 | 3265 | **1938** | **1938** | 1937 | 1931 | 1934 |
| p-hat700-1 | 700 | 60999 | 33413 | 31856 | 33426 | 33440 | **33450** |
| email-univ | 1133 | 5451 | **3775** | 3764 | 3657 | 3765 | 3736 |

The results indicate that X2GNN achieves an average gap of 0.12% from the best solutions among the compared ones under the 64x256 setting, which has a maximum running time of 11 seconds on the largest instance with $10^4$ nodes. ANYCSP under the 64x256 setting has a similar running time and achieves an average gap of 0.69% from the best solution. When we increase the number of refinement iterations by one order of magnitude, the average gaps achieved by X2GNN and ANYCSP are 0.12% and 0.56%, respectively. Despite having an order of magnitude longer runtime, ANYCSP achieves a higher average gap than X2GNN at even a short runtime. These results show that both methods are scalable and can generate high-quality solutions to larger and different distributions than those seen during training; however, X2GNN performs better than ANYCSP.

Additionally, X2GNN generates solutions with a smaller gap on average than the second best traditional heuristic KHLWG, whereas ANYCSP consistently generates solutions with higher gap than KHLWG.

Table 12: Results for Max Cut on Gset dataset, presenting the cut size for each instance. The last row shows the average gap in percentages from the best solution.

| Graph | BLS | DSDP | KHLWG | X2GNN 64x256 | ANYCSP 64x256 | X2GNN 64x2560 | ANYCSP 64x2560 |
|---|---|---|---|---|---|---|---|
| G1 | 11624 | - | 11624 | 11583 | 11582 | 11598 | 11616 |
| G14 | 3064 | 2922 | 3061 | 3054 | 3021 | 3058 | 3035 |
| G15 | 3050 | 2938 | 3050 | 3043 | 3011 | 3036 | 3024 |
| G16 | 3052 | - | 3052 | 3040 | 3011 | 3045 | 3024 |
| G17 | 3047 | - | 3046 | 3037 | 3002 | 3039 | 3018 |
| G2 | 11620 | - | 11620 | 11575 | 11585 | 11588 | 11601 |
| G22 | 13359 | 12960 | 13359 | 13339 | 13263 | 13347 | 13285 |
| G23 | 13344 | 13006 | 13342 | 13319 | 13247 | 13329 | 13309 |
| G24 | 13377 | 12933 | 13337 | 13310 | 13273 | 13319 | 13276 |
| G25 | 13340 | - | 13332 | 13311 | 13248 | 13327 | 13287 |
| G26 | 13328 | - | 13328 | 13296 | 13236 | 13306 | 13272 |
| G3 | 11622 | - | 11620 | 11583 | 11581 | 11591 | 11614 |
| G35 | 7684 | - | 7678 | 7648 | 7543 | 7646 | 7575 |
| G36 | 7678 | - | 7670 | 7646 | 7529 | 7654 | 7566 |
| G37 | 7689 | - | 7682 | 7665 | 7546 | 7666 | 7593 |
| G38 | 7687 | - | 7683 | 7649 | 7547 | 7658 | 7570 |
| G4 | 11646 | - | 11646 | 11597 | 11612 | 11617 | 11633 |
| G43 | 6660 | - | 6660 | 6655 | 6622 | 6659 | 6651 |
| G44 | 6650 | - | 6639 | 6645 | 6622 | 6646 | 6631 |
| G45 | 6654 | - | 6652 | 6640 | 6625 | 6652 | 6641 |
| G46 | 6649 | - | 6649 | 6639 | 6622 | 6644 | 6637 |
| G47 | 6657 | - | 6665 | 6651 | 6626 | 6654 | 6647 |
| G48 | 6000 | 6000 | 6000 | 6000 | 6000 | 6000 | 6000 |
| G49 | 6000 | 6000 | 6000 | 6000 | 6000 | 6000 | 6000 |
| G5 | 11631 | - | 11631 | 11596 | 11583 | 11594 | 11623 |
| G50 | 5880 | 5880 | 5880 | 5872 | 5846 | 5880 | 5876 |
| G51 | 3848 | - | 3847 | 3838 | 3791 | 3837 | 3810 |
| G52 | 3851 | - | 3849 | 3835 | 3792 | 3840 | 3805 |
| G53 | 3850 | - | 3848 | 3841 | 3796 | 3844 | 3812 |
| G54 | 3852 | - | 3851 | 3842 | 3791 | 3841 | 3802 |
| G55 | 10294 | 9960 | 10236 | 10241 | 10066 | 10283 | 10141 |
| G58 | 19263 | - | 19248 | 19167 | 18843 | 19186 | 18937 |
| G60 | 14176 | 13610 | 14057 | 14121 | 13855 | 14146 | 13952 |
| G63 | 26997 | 8017 | 26963 | 26863 | 26405 | 26905 | 26512 |
| G70 | 9541 | 9456 | 9458 | 9492 | 9189 | 9525 | 9290 |
| GeoMean | 7726.5 | - | 7718.6 | 7703.5 | 7638.6 | 7710.3 | 7667.0 |
| Mean Gap | 0.01% | - | 0.13% | 0.12% | 0.69% | 0.11% | 0.56% |

