# OpenReview forum: "Learning to Explore and Exploit with GNNs for Unsupervised Combinatorial Optimization"
_ICLR.cc/2025/Conference — ICLR 2025 Poster_

### Official Review · Reviewer_aRXC · 2024-10-27

**Soundness:** 2
**Presentation:** 2
**Contribution:** 2
**Rating:** 5
**Confidence:** 4

**Summary:**

This paper proposes GNN-based framework to solve several classic combinatorial optimization problems. The proposed approach behaves like a population-based heuristic method. Since extensive efforts have been devoted to the development of machine learning methods for addressing combinatorial optimization, I'm concerned about whether it can outperform the state-of-the-art algorithms.

**Strengths:**

- The proposed network generates $K$-coupled solutions and behaves like a population-based heuristic method. This is kind of novel.

**Weaknesses:**

I have a few concerns below.

- Line 245, the loss function includes constraint satisfaction and solution diversity. How to choose $\lambda_1$ and $\lambda_2$? Usually, the penalty method demonstrate very weak generalization capabilities. Hence, I personally think combining several terms in the loss function is not a good idea.

- For MCut and MIS comparison, a state-of-the-art algorithm [1] should be considered as baseline. This algorithm [1] is quite scalable and able to provide high-quality solutions to MCut and MIS.

- The proposed algorithm is not very scalable. For example, in Table 2 and 3, the computational time increases quickly with the problem size. MIS, MC and MCut are simple combinatorial optimization problems. Why not consider some large-sized instances (for example, Gset instances for MCut,  https://web.stanford.edu/~yyye/yyye/Gset)? How does the proposed algorithm perform?

[1] Schuetz, M.J., Brubaker, J.K. and Katzgraber, H.G., 2022. Combinatorial optimization with physics-inspired graph neural networks. Nature Machine Intelligence, 4(4), pp.367-377.

**Questions:**

See the weakness part.

---

> ### Author Response · Authors · 2024-11-21
> **Review 4**
>
> Thank you for your helpful review and suggestions. We address individual points below:
>
>
> ## 1 Penalty terms impeding generalization
> > Line 245, the loss function includes constraint satisfaction and solution diversity. How to choose $\lambda_1$ and $\lambda_2$ Usually, the penalty method demonstrate very weak generalization capabilities. Hence, I personally think combining several terms in the loss function is not a good idea.
>
> This is a good question, empirically we set $\lambda_1$ the constraint coefficient to be 1, and $\lambda_2$ the diversity coefficient to be 0.75 for each problem. We set these initially for all problems and didn’t tune them, but there may be more performant coefficients. Overall, our reasoning for setting these coefficients was that $\lambda_1$ needs to be greater than or equal to 1 to ensure that the model can’t improve solution quality by generating infeasible solutions. And $\lambda_2$ needs to be smaller to prevent the diversity loss from overtaking the optimization-specific loss terms of objective and constraint satisfaction. We will make this clear to help others set these hyperparameters in an intuitive manner. We do find that setting these parameters in a reasonable manner empirically encourages generalization to unseen, larger, and out-of-distribution instances.
>
> ## 2 additional baseline
> > For MCut and MIS comparison, a state-of-the-art algorithm [1] should be considered as baseline. This algorithm [1] is quite scalable and able to provide high-quality solutions to MCut and MIS.
>
> Thank you for pointing out this previous work. We ran new experiments comparing X2GNN and this approach finding that X2GNN greatly outperforms the previous state of the art in both runtime and speed for Max Cut instances on GSet (Appendix Table 12) and independent set on regular graphs (Appendix Figure 4). Furthermore, we found that X2GNN can scale to much larger instances, providing higher-quality solutions across the board and having much lower runtime. Furthermore, X2GNN scales in memory usage as well, in that the previous work runs into memory limitations for d-regular graphs with $10^6$ nodes since it uses a penalty matrix with $N^2$ entries. We provide these experimental results in the appendix.
>
>
> ## 3 GSet instances
> > The proposed algorithm is not very scalable. For example, in Table 2 and 3, the computational time increases quickly with the problem size. MIS, MC and MCut are simple combinatorial optimization problems. Why not consider some large-sized instances (for example, Gset instances for MCut, https://web.stanford.edu/~yyye/yyye/Gset)? How does the proposed algorithm perform?
>
> Thank you for pointing us to these instances to investigate the scalability of X2GNN which has greatly improved our understanding of X2GNN’s scalability. We conducted new experiments on large regular graphs (Appendix Figure 4) for MIS and on Gset for MCut (Appendix Table 12). Our algorithm is able to scale to graphs with millions of nodes and generates solutions in 500 seconds. For the Gset, our algorithm is able to outperform the suggested baseline by a large margin, achieving an average gap of 0.09% from the best-known whereas the PI-GNN achieves an average gap of 1.39%. Furthermore, we additionally demonstrate that X2GNN outperforms KaMIS, an MIS-specific metaheuristic, on the largest instances while also using less time.
>
> ## References
>
> [1] Schuetz, M.J., Brubaker, J.K. and Katzgraber, H.G., 2022. Combinatorial optimization with physics-inspired graph neural networks. Nature Machine Intelligence, 4(4), pp.367-377.

---

> > ### Comment · Reviewer_aRXC · 2024-12-02
> >
> > Thanks for your efforts for addressing my comments. I still have a few questions.
> > 1. The penalization parameter is still an issue not fully addressed.
> >
> > 2. I noticed that you include ANYCSP in Table 11 but not in Table 12, and include PI-GNN in Table 12 but not in Table 11. Can you clarify? In my experience, both are very good methods for addressing Max-Cut.
> >
> > 3. My previous comment "The proposed algorithm is not very scalable". "Scalable" means "large-sized instances", hence why not consider all the 10,000-node Gset instances in Table 12? Why not report the solution time for each method?

---

> > > ### Author Response · Authors · 2024-12-02
> > > **Part 1**
> > >
> > > Thank you for your follow-up questions.  Below, we address the remaining points with clarifications and additional experiments:
> > >
> > > **1. Penalization Parameters**
> > > > The penalization parameter is still an issue not fully addressed.
> > >
> > > For the penalization parameters, we will explain in writing how we obtained our parameters so that others may readily deploy it to their settings. We will also clarify that the parameter values were chosen simply based on intuition for what the parameters represent and were not tuned in any way based on performance on the data. We found that using this simple technique X2GNN was still able to outperform learning based approaches and several traditional OR approaches across many settings. We agree that tuning hyper-parameters may be tricky, but we found that we can have a performant learned heuristic that generalizes well to instances that are larger and out-of-distribution without any parameter tuning by solely setting the parameters in a reasonable way.
> > >
> > > To provide further clarity. Empirically, we set $\lambda_1$, the constraint coefficient, to 1, and $\lambda_2$, the diversity coefficient, to 0.75. These values were initially set for all problems and were not tuned further, though there may indeed be alternative coefficients for specific cases that enable faster convergence or convergence to better models. Our reasoning for choosing these coefficients was that $\lambda_1$ needs to be at least 1 to ensure that the model cannot improve solution quality by generating infeasible solutions. Meanwhile, $\lambda_2$ was set to a smaller value to prevent the diversity loss from overshadowing the optimization-specific losses related to objective and constraint satisfaction.
> > >
> > > Please let us know if there is something we can better clarify or demonstrate about the penalization parameters, and we will make our best effort to do so before the end of the comment period.

---

> > > ### Author Response · Authors · 2024-12-02
> > > **Part 2**
> > >
> > > **Tables for GSET performance**
> > >
> > > To address questions 2 and 3 we run more comprehensive evaluation on all unweighted GSET instances, including more problem instances and more baselines. We include the tables here, table 13 and 14.
> > >
> > > > Table 13. Results for Max Cut on Gset dataset, presenting the cut size for each instance. The last row shows the average gap in percentages from the best solution. The best results among the learning-based methods are shown in bold. The global best results among all methods are shown in italic.
> > >
> > > | Graph | n   | m   | BLS     	| DSDP    	| KHLWG   	| X2GNN 64x256  | ANYCSP 64x256  | X2GNN 64x2560  | ANYCSP 64x2560  |
> > > | :---- | :-- | :-- | :----------: | :----------: | :----------: | :------------: | :-------------: | :-------------: | :-------------: |
> > > | G1 | 800 | 19176 | *11624* | - | *11624* | 11583 | 11598 | 11582 | **11616** |
> > > | G14 | 800 | 4694 | *3064* | 2922 | 3061 | 3054 | **3058** | 3021 | 3035 |
> > > | G15 | 800 | 4661 | *3050* | 2938 | *3050* | 3043 | **3046** | 3011 | 3024 |
> > > | G16 | 800 | 4672 | *3052* | - | *3052* | 3040 | **3045** | 3011 | 3024 |
> > > | G17 | 800 | 4667 | *3047* | - | 3046 | 3037 | **3039** | 3002 | 3018 |
> > > | G2 | 800 | 19176 | *11620* | - | *11620* | 11575 | 11588 | 11585 | **11601** |
> > > | G22 | 2000 | 19990 | *13359* | 12960 | *13359* | 13339 | **13347** | 13263 | 13285 |
> > > | G23 | 2000 | 19990 | *13344* | 13006 | 13342 | 13319 | **13329** | 13247 | 13309 |
> > > | G24 | 2000 | 19990 | *13377* | 12933 | 13337 | 13310 | **13319** | 13273 | 13276 |
> > > | G25 | 2000 | 19990 | *13340* | - | 13332 | 13311 | **13327** | 13248 | 13287 |
> > > | G26 | 2000 | 19990 | *13328* | - | *13328* | 13296 | **13306** | 13236 | 13272 |
> > > | G3 | 800 | 19176 | *11622* | - | 11620 | 11583 | 11591 | 11581 | **11614** |
> > > | G35 | 2000 | 11778 | *7684* | - | 7678 | **7648** | 7646 | 7543 | 7575 |
> > > | G36 | 2000 | 11766 | *7678* | - | 7670 | 7646 | **7654** | 7529 | 7566 |
> > > | G37 | 2000 | 11785 | *7689* | - | 7682 | 7665 | **7666** | 7546 | 7593 |
> > > | G38 | 2000 | 11779 | *7687* | - | 7683 | 7649 | **7658** | 7547 | 7570 |
> > > | G4 | 800 | 19176 | *11646* | - | *11646* | 11597 | 11617 | 11612 | **11633** |
> > > | G43 | 1000 | 9990 | *6660* | - | *6660* | 6655 | **6659** | 6622 | 6651 |
> > > | G44 | 1000 | 9990 | *6650* | - | 6639 | 6645 | **6646** | 6622 | 6631 |
> > > | G45 | 1000 | 9990 | *6654* | - | 6652 | 6640 | **6652** | 6625 | 6641 |
> > > | G46 | 1000 | 9990 | *6649* | - | *6649* | 6639 | **6644** | 6622 | 6637 |
> > > | G47 | 1000 | 9990 | 6657 | - | *6665* | 6651 | **6654** | 6626 | 6647 |
> > > | G48 | 3000 | 6000 | *6000* | *6000* | *6000* | ***6000*** | ***6000*** | ***6000*** | ***6000*** |
> > > | G49 | 3000 | 6000 | *6000* | *6000* | *6000* | ***6000*** | ***6000*** | ***6000*** | ***6000*** |
> > > | G5 | 800 | 19176 | *11631* | - | *11631* | 11596 | 11594 | 11583 | **11623** |
> > > | G50 | 3000 | 6000 | *5880* | *5880* | *5880* | 5872 | ***5880*** | 5846 | 5876 |
> > > | G51 | 1000 | 5909 | *3848* | - | 3847 | **3838** | 3837 | 3791 | 3810 |
> > > | G52 | 1000 | 5916 | *3851* | - | 3849 | 3835 | **3840** | 3792 | 3805 |
> > > | G53 | 1000 | 5914 | *3850* | - | 3848 | 3841 | **3844** | 3796 | 3812 |
> > > | G54 | 1000 | 5916 | *3852* | - | 3851 | **3842** | 3841 | 3791 | 3802 |
> > > | G55 | 5000 | 12498 | *10294* | 9960 | 10236 | 10241 | **10283** | 10066 | 10141 |
> > > | G58 | 5000 | 29570 | *19263* | - | 19248 | 19167 | **19186** | 18843 | 18937 |
> > > | G60 | 7000 | 17148 | *14176* | 13610 | 14057 | 14121 | **14146** | 13855 | 13952 |
> > > | G63 | 7000 | 41459 | *26997* | 8017 | 26963 | 26863 | **26905** | 26405 | 26512 |
> > > | G70 | 10000 | 9999 | *9541* | 9456 | 9458 | 9492 | **9525** | 9189 | 9290 |
> > > | MeanGap | - | - | 0.01% | - | 0.13% | 0.12% | 0.11% | 0.69% | 0.56%|

---

> > > > ### Comment · Reviewer_aRXC · 2024-12-02
> > > >
> > > > Where are the results for G67, G72, G77, G81? As far as I know, there are 5 10000-node instances and only G70 is included.

---

> > > > > ### Author Response · Authors · 2024-12-03
> > > > >
> > > > > We include results only for unweighted instances, as the other four are weighted problem instances. Currently, X2GNN is primarily designed for unweighted instances, and we plan to explore approaches tailored to weighted problems in future work.

---

> > > > > > ### Comment · Reviewer_aRXC · 2024-12-03
> > > > > >
> > > > > > Thanks for your reply. As I'm kind of familiar with Max-Cut instances, I still have a few questions.
> > > > > >
> > > > > > 1. Do you consider Gset graph with edge weights of 1, -1 as unweighted? Could you be more specific about which types of Max-Cut instances your approach is applicable to?
> > > > > >
> > > > > > 2. From Table 13 and 14, it seems to me that ANYCSP outperforms the proposed approach since ANYCSP 64x256 leads to the smallest gap, but the authors claimed "X2GNN slightly outperforms ANYCSP, with both being scalable achieving high-quality solutions in a reasonable amount of time". Did I misunderstand something?
> > > > > >
> > > > > > 3. I noticed that in Table 12 and Table 13, there are different versions of your approaches, ($X^2GNN 256x32$, and $X^2 GNN 64x256$). Could you please clarify?

---

> > > > > > > ### Author Response · Authors · 2024-12-03
> > > > > > >
> > > > > > > > Thank you for your prompt and detailed feedback. We sincerely apologize for the mistakes in the column names in Tables 13 and 14, which may have caused confusion. We appreciate you bringing these issues to our attention, as it allows us to clarify and address them below.
> > > > > > >
> > > > > > > **1. Edge Weights**
> > > > > > >
> > > > > > > Regarding Gset Graphs: We classify Gset graphs with edge weights of 1 and -1 as weighted and, therefore, do not include them in Tables 13 and 14. These tables are limited to unweighted graphs or graphs where all edge weights are 1. Our current trained models are designed exclusively for unweighted graphs, so we did not test them on graphs with weights of 1 and -1.
> > > > > > >
> > > > > > > That said, extending this approach to weighted Max Cut problems is straightforward. This can be achieved by modifying the Graph Isomorphism Network to incorporate a GNN architecture that supports edge features. As a proof of concept, we trained a model using X2GNN for the node-weighted Maximum Independent Set problem during rebuttal, demonstrating X2GNN’s capability to address weighted problem instances. A similar modification, combined with an updated loss function to account for edge weights, would enable training a model for the weighted Max-Cut problem.
> > > > > > >
> > > > > > >
> > > > > > >
> > > > > > > **2 & 3**
> > > > > > >
> > > > > > > We acknowledge an error in the column labels, and we sincerely apologize for this mistake. Specifically, we inadvertently swapped the column names of X2GNN 64x2560 and ANYCSP 64x256.
> > > > > > > Initially, we grouped methods by similar runtime (placing shorter runtimes together), but later decided that grouping them by method would be more intuitive. Additionally, as you correctly observed, we typically report results for x32 versions of X2GNN rather than x64 versions. However, for this comparison, we used the x64 version of ANYCSP to ensure comparable runtimes across methods. Unfortunately, the columns were mislabeled as 64x256 and 64x2560 instead of the correct labels: 32x256 and 32x2560.
> > > > > > >
> > > > > > > We sincerely regret any confusion caused by these errors and thank you again for bringing them to our attention.
> > > > > > >
> > > > > > > The corrected column labels for Table 13 should be: **Graph, n, m, BLS, DSDP, KHLWG, X2GNN 32x256, X2GNN 32x2560, ANYCSP 64x256, ANYCSP 64x2560**.
> > > > > > >
> > > > > > > For Table 14, the corrected labels are: **Graph, n, m, BLS, X2GNN 32x256, X2GNN 32x2560, ANYCSP 64x256, ANYCSP 64x2560**.
> > > > > > >
> > > > > > > We post the tables with the correct column names below for your reference.

---

> > > > > > > ### Author Response · Authors · 2024-12-03
> > > > > > > **Table 13 with correct column names**
> > > > > > >
> > > > > > > | Graph | n   | m   | BLS     	| DSDP    	| KHLWG   	| X2GNN 32x256  |  X2GNN 32x2560  | ANYCSP 64x256  | ANYCSP 64x2560  |
> > > > > > > | :---- | :-- | :-- | :----------: | :----------: | :----------: | :------------: | :-------------: | :-------------: | :-------------: |
> > > > > > > | G1 | 800 | 19176 | *11624* | - | *11624* | 11583 | 11598 | 11582 | **11616** |
> > > > > > > | G14 | 800 | 4694 | *3064* | 2922 | 3061 | 3054 | **3058** | 3021 | 3035 |
> > > > > > > | G15 | 800 | 4661 | *3050* | 2938 | *3050* | 3043 | **3046** | 3011 | 3024 |
> > > > > > > | G16 | 800 | 4672 | *3052* | - | *3052* | 3040 | **3045** | 3011 | 3024 |
> > > > > > > | G17 | 800 | 4667 | *3047* | - | 3046 | 3037 | **3039** | 3002 | 3018 |
> > > > > > > | G2 | 800 | 19176 | *11620* | - | *11620* | 11575 | 11588 | 11585 | **11601** |
> > > > > > > | G22 | 2000 | 19990 | *13359* | 12960 | *13359* | 13339 | **13347** | 13263 | 13285 |
> > > > > > > | G23 | 2000 | 19990 | *13344* | 13006 | 13342 | 13319 | **13329** | 13247 | 13309 |
> > > > > > > | G24 | 2000 | 19990 | *13377* | 12933 | 13337 | 13310 | **13319** | 13273 | 13276 |
> > > > > > > | G25 | 2000 | 19990 | *13340* | - | 13332 | 13311 | **13327** | 13248 | 13287 |
> > > > > > > | G26 | 2000 | 19990 | *13328* | - | *13328* | 13296 | **13306** | 13236 | 13272 |
> > > > > > > | G3 | 800 | 19176 | *11622* | - | 11620 | 11583 | 11591 | 11581 | **11614** |
> > > > > > > | G35 | 2000 | 11778 | *7684* | - | 7678 | **7648** | 7646 | 7543 | 7575 |
> > > > > > > | G36 | 2000 | 11766 | *7678* | - | 7670 | 7646 | **7654** | 7529 | 7566 |
> > > > > > > | G37 | 2000 | 11785 | *7689* | - | 7682 | 7665 | **7666** | 7546 | 7593 |
> > > > > > > | G38 | 2000 | 11779 | *7687* | - | 7683 | 7649 | **7658** | 7547 | 7570 |
> > > > > > > | G4 | 800 | 19176 | *11646* | - | *11646* | 11597 | 11617 | 11612 | **11633** |
> > > > > > > | G43 | 1000 | 9990 | *6660* | - | *6660* | 6655 | **6659** | 6622 | 6651 |
> > > > > > > | G44 | 1000 | 9990 | *6650* | - | 6639 | 6645 | **6646** | 6622 | 6631 |
> > > > > > > | G45 | 1000 | 9990 | *6654* | - | 6652 | 6640 | **6652** | 6625 | 6641 |
> > > > > > > | G46 | 1000 | 9990 | *6649* | - | *6649* | 6639 | **6644** | 6622 | 6637 |
> > > > > > > | G47 | 1000 | 9990 | 6657 | - | *6665* | 6651 | **6654** | 6626 | 6647 |
> > > > > > > | G48 | 3000 | 6000 | *6000* | *6000* | *6000* | ***6000*** | ***6000*** | ***6000*** | ***6000*** |
> > > > > > > | G49 | 3000 | 6000 | *6000* | *6000* | *6000* | ***6000*** | ***6000*** | ***6000*** | ***6000*** |
> > > > > > > | G5 | 800 | 19176 | *11631* | - | *11631* | 11596 | 11594 | 11583 | **11623** |
> > > > > > > | G50 | 3000 | 6000 | *5880* | *5880* | *5880* | 5872 | ***5880*** | 5846 | 5876 |
> > > > > > > | G51 | 1000 | 5909 | *3848* | - | 3847 | **3838** | 3837 | 3791 | 3810 |
> > > > > > > | G52 | 1000 | 5916 | *3851* | - | 3849 | 3835 | **3840** | 3792 | 3805 |
> > > > > > > | G53 | 1000 | 5914 | *3850* | - | 3848 | 3841 | **3844** | 3796 | 3812 |
> > > > > > > | G54 | 1000 | 5916 | *3852* | - | 3851 | **3842** | 3841 | 3791 | 3802 |
> > > > > > > | G55 | 5000 | 12498 | *10294* | 9960 | 10236 | 10241 | **10283** | 10066 | 10141 |
> > > > > > > | G58 | 5000 | 29570 | *19263* | - | 19248 | 19167 | **19186** | 18843 | 18937 |
> > > > > > > | G60 | 7000 | 17148 | *14176* | 13610 | 14057 | 14121 | **14146** | 13855 | 13952 |
> > > > > > > | G63 | 7000 | 41459 | *26997* | 8017 | 26963 | 26863 | **26905** | 26405 | 26512 |
> > > > > > > | G70 | 10000 | 9999 | *9541* | 9456 | 9458 | 9492 | **9525** | 9189 | 9290 |
> > > > > > > | MeanGap | - | - | 0.01% | - | 0.13% | 0.12% | 0.11% | 0.69% | 0.56%|

---

> > > > > > > ### Author Response · Authors · 2024-12-03
> > > > > > >
> > > > > > > | Graph | n| m |  BLS | X2GNN 32x256| X2GNN 32x2560 | ANYCSP 64x256 | ANYCSP 64x2560 |
> > > > > > > | :---- | :-- | :-- | :----------: | :----------: | :----------: | :------------: | :-------------: |
> > > > > > > | G1 | 800 | 19176 | 260 | 5.3 | 43.3 | 5.2 | 51.9 |
> > > > > > > | G14 | 800 | 4694 | 2380 | 2.9 | 30.4 | 2.1 | 20.6 |
> > > > > > > | G15 | 800 | 4661 | 860 | 2.9 | 30.4 | 2.0 | 20.4 |
> > > > > > > | G16 | 800 | 4672 | 1400 | 2.9 | 30.2 | 2.0 | 20.4 |
> > > > > > > | G17 | 800 | 4667 | 1920 | 2.9 | 30.4 | 2.0 | 20.4 |
> > > > > > > | G2 | 800 | 19176 | 820 | 4.1 | 41.9 | 5.2 | 51.9 |
> > > > > > > | G22 | 2000 | 19990 | 11200 | 5.1 | 51.2 | 6.4 | 64.2 |
> > > > > > > | G23 | 2000 | 19990 | 5560 | 5.1 | 51.0 | 6.4 | 64.1 |
> > > > > > > | G24 | 2000 | 19990 | 6220 | 5.1 | 51.0 | 6.4 | 64.1 |
> > > > > > > | G25 | 2000 | 19990 | 2960 | 5.0 | 51.0 | 6.4 | 64.1 |
> > > > > > > | G26 | 2000 | 19990 | 8580 | 5.1 | 50.9 | 6.4 | 64.1 |
> > > > > > > | G3 | 800 | 19176 | 1660 | 4.1 | 42.0 | 5.2 | 51.9 |
> > > > > > > | G35 | 2000 | 11778 | 8840 | 4.4 | 44.3 | 4.6 | 45.9 |
> > > > > > > | G36 | 2000 | 11766 | 12080 | 4.4 | 44.3 | 4.6 | 45.9 |
> > > > > > > | G37 | 2000 | 11785 | 8880 | 4.4 | 44.1 | 4.6 | 46.0 |
> > > > > > > | G38 | 2000 | 11779 | 9220 | 4.4 | 44.3 | 4.6 | 46.0 |
> > > > > > > | G4 | 800 | 19176 | 4280 | 4.1 | 41.9 | 5.2 | 51.9 |
> > > > > > > | G43 | 1000 | 9990 | 520 | 3.5 | 35.7 | 3.4 | 33.5 |
> > > > > > > | G44 | 1000 | 9990 | 860 | 3.5 | 35.7 | 3.4 | 33.5 |
> > > > > > > | G45 | 1000 | 9990 | 2080 | 3.5 | 35.6 | 3.4 | 33.5 |
> > > > > > > | G46 | 1000 | 9990 | 1340 | 3.5 | 35.6 | 3.4 | 33.5 |
> > > > > > > | G47 | 1000 | 9990 | 2040 | 3.5 | 35.6 | 3.4 | 33.5 |
> > > > > > > | G48 | 3000 | 6000 | 0 | 4.8 | 48.8 | 3.3 | 3.6 |
> > > > > > > | G49 | 3000 | 6000 | 0 | 4.8 | 48.7 | 3.6 | 2.6 |
> > > > > > > | G5 | 800 | 19176 | 280 | 4.1 | 41.9 | 5.2 | 51.9 |
> > > > > > > | G50 | 3000 | 6000 | 3380 | 4.8 | 48.7 | 4.2 | 42.1 |
> > > > > > > | G51 | 1000 | 5909 | 1620 | 3.1 | 32.3 | 2.5 | 24.8 |
> > > > > > > | G52 | 1000 | 5916 | 1560 | 3.1 | 32.2 | 2.5 | 24.8 |
> > > > > > > | G53 | 1000 | 5914 | 2340 | 3.1 | 32.2 | 2.5 | 24.8 |
> > > > > > > | G54 | 1000 | 5916 | 2620 | 3.1 | 32.3 | 2.5 | 24.8 |
> > > > > > > | G55 | 5000 | 12498 | 16840 | 7.1 | 71.5 | 7.3 | 73.1 |
> > > > > > > | G58 | 5000 | 29570 | 27080 | 8.6 | 84.6 | 11.2 | 111.7 |
> > > > > > > | G60 | 7000 | 17148 | 56440 | 9.2 | 91.5 | 10.1 | 100.8 |
> > > > > > > | G63 | 7000 | 41459 | 126360 | 11.2 | 111.0 | 16.1 | 160.9 |
> > > > > > > | G70 | 10000 | 9999 | 227300 | 11.2 | 109.7 | 11.1 | 111.3 |

---

> > > ### Author Response · Authors · 2024-12-02
> > > **Part 4**
> > >
> > > **2. Inclusion of ANYCSP and PI-GNN in Tables 11 and 12**
> > > > I noticed that you include ANYCSP in Table 11 but not in Table 12, and include PI-GNN in Table 12 but not in Table 11. Can you clarify? In my experience, both are very good methods for addressing Max-Cut.
> > >
> > >
> > > Thank you for pointing out the oversight in Tables 11 and 12 regarding the inclusion of ANYCSP and PI-GNN. We obtain results for ANYCSP on all GSET max cut instances in a new table, Table 13, which we include below. We discuss the results in response to question 3, in that X2GNN slightly outperforms ANYCSP, with both being scalable achieving high-quality solutions in a reasonable amount of time.
> > >
> > > However, PI-GNN does not provide code for max cut, so we cannot include results on instances in Table 11. The solution quality results we give for PI-GNN are obtained from the previous work. Nevertheless, we believe that our results demonstrate that X2GNN outperforms PI-GNN in solution quality for max cut. Additionally, we found that there is one instance that should be "easy" to solve, GSET instance G49. This instance is a bipartite graph as it has 3,000 nodes, and 6,000 edges, and there exists a cut that cuts all 6,000 edges (obtained by all approaches except PI-GNN). Note that solving max cut on bipartite graphs is trivial as one simply needs to identify one side of the bipartite graph in order to cut all edges. The fact that X2GNN obtains the optimal solution here (cutting all edges) while PI-GNN fails to obtain optimality is promising in that X2GNN performs reasonably when it is given a problem that should be easy to solve given the right algorithm. Regardless, X2GNN outperforms PI-GNN in all problem instances on which PI-GNN is evaluated.

---

> > > ### Author Response · Authors · 2024-12-02
> > > **Part 5**
> > >
> > > **3. Scalability and Inclusion of All Gset Instances**
> > > > My previous comment "The proposed algorithm is not very scalable". "Scalable" means "large-sized instances", hence why not consider all the 10,000-node Gset instances in Table 12? Why not report the solution time for each method?
> > >
> > > Regarding scalability, we provide additional results for X2GNN on all unweighted GSET instances, demonstrating that it can generalize to more large instances. We report results comparing to the instances PI-GNN compares against and unfortunately, there is no public implementation of PI-GNN for max cut. This means we cannot get reliable runtime estimates using the same hardware. However, we do evaluate X2GNN’s scalability against PI-GNN and KaMIS on MIS for d-regular graphs showing that X2GNN greatly improves runtime and solution quality compared to PI-GNN. Additionally, X2GNN finds better-quality solutions than KaMIS for a similar runtime for the largest problem instances suggesting that X2GNN is highly scalable.
> > >
> > > ### Solution quality
> > > Table 13 shows the cut size achieved by the traditional heuristics BLS, DSDP, KHLWG and learning based heuristics X2GNN and ANYCSP for each instance. To aggregate the results, we first compute the virtual best solution by taking the maximum value found by any of the compared algorithms. Then, for each instance we compute the average gap from the best as 1- solution/best and take the mean to calculate the average gap from the virtual best solution.
> > >
> > > The results indicate that X2GNN achieves an average gap of 0.12% from the best solutions among the compared ones under the 64x256 setting, which has a maximum running time of 11 seconds on the largest instance with 10^4 nodes. ANYCSP under  the 64x256 setting has a similar running time and achieves an average gap of 0.69% from the best solution. When we increase the number of refinement iterations by one order of magnitude, the average gaps achieved by X2GNN and ANYCSP are 0.12% and 0.56% respectiely. Despite having an order of magnitude longer runtime, ANYCSP achieves a higher average gap than X2GNN at even a short runtime. These results show that both methods are scalable and can generate high quality solutions to larger and different distributions than those seen during training; however, X2GNN performs better than ANYCSP.
> > >
> > > Additionally, X2GNN generates solutions with a smaller gap on average than the second best traditional heuristic KHLWG, whereas ANYCSP consistently generates solutions with higher gap than KHLWG.
> > >
> > > ### Runtime Comparisons
> > > Table 14 shows the running time (in seconds) of each method on each instance. The running times of X2GNN and ANYCSP are measured on the same machine as we have the public implementation. Since we do not have the implementation of BLS, we use the running times reported in [1]. We note that these runtimes were calculated on an older machine and used only one CPU core per instance. Thus, the comparison between BLS and the learning-based heuristics are meant to give a broad perspective of runtime but machine details should be considered.
> > >
> > > The running time of X2GNN and ANYCSP are similar. On a large graph with 10^4 nodes both methods roughly takes 11 seconds when 64 solutions are refined over 259 steps. Both algorithms running time is linear with respect to the refinement steps. Both take roughly 110 seconds when 64 solutions are refined over 2559 steps. The best traditional heuristic BLS takes around 227,300 seconds (roughly 63 hours). Still, the running time of BLS can be about 3 or 4 orders of magnitude larger than the learning-based heuristic.
> > >
> > >
> > > [1] Benlic, Una, and Jin-Kao Hao. "Breakout local search for the max-cutproblem." Engineering Applications of Artificial Intelligence 26, no. 3 (2013): 1162-1173.

---

> ### Comment · Area_Chair_PbN1 · 2024-11-25
>
> Dear Reviewer,
>
> This is a kind reminder that the dicussion phase will be ending soon on November 26th. Please read the author's responses and engage in a constructive discussion with the authors.
>
> Thank you for your time and cooperation.
>
> Best,
>
> Area Chair

---

> ### Author Response · Authors · 2024-12-02
> **Part 3**
>
> > Table 14. The running times for Max Cut on Gset dataset, presenting the run time in seconds  for each instance. The running times of BLS are taken from [1] that uses  a different machine as there is no implementation released for BLS.
>
> | Graph | n| m |  BLS | X2GNN 64x256| X2GNN 64x2560 | ANYCSP 64x256 | ANYCSP 64x2560 |
> | :---- | :-- | :-- | :----------: | :----------: | :----------: | :------------: | :-------------: |
> | G1 | 800 | 19176 | 260 | 5.3 | 43.3 | 5.2 | 51.9 |
> | G14 | 800 | 4694 | 2380 | 2.9 | 30.4 | 2.1 | 20.6 |
> | G15 | 800 | 4661 | 860 | 2.9 | 30.4 | 2.0 | 20.4 |
> | G16 | 800 | 4672 | 1400 | 2.9 | 30.2 | 2.0 | 20.4 |
> | G17 | 800 | 4667 | 1920 | 2.9 | 30.4 | 2.0 | 20.4 |
> | G2 | 800 | 19176 | 820 | 4.1 | 41.9 | 5.2 | 51.9 |
> | G22 | 2000 | 19990 | 11200 | 5.1 | 51.2 | 6.4 | 64.2 |
> | G23 | 2000 | 19990 | 5560 | 5.1 | 51.0 | 6.4 | 64.1 |
> | G24 | 2000 | 19990 | 6220 | 5.1 | 51.0 | 6.4 | 64.1 |
> | G25 | 2000 | 19990 | 2960 | 5.0 | 51.0 | 6.4 | 64.1 |
> | G26 | 2000 | 19990 | 8580 | 5.1 | 50.9 | 6.4 | 64.1 |
> | G3 | 800 | 19176 | 1660 | 4.1 | 42.0 | 5.2 | 51.9 |
> | G35 | 2000 | 11778 | 8840 | 4.4 | 44.3 | 4.6 | 45.9 |
> | G36 | 2000 | 11766 | 12080 | 4.4 | 44.3 | 4.6 | 45.9 |
> | G37 | 2000 | 11785 | 8880 | 4.4 | 44.1 | 4.6 | 46.0 |
> | G38 | 2000 | 11779 | 9220 | 4.4 | 44.3 | 4.6 | 46.0 |
> | G4 | 800 | 19176 | 4280 | 4.1 | 41.9 | 5.2 | 51.9 |
> | G43 | 1000 | 9990 | 520 | 3.5 | 35.7 | 3.4 | 33.5 |
> | G44 | 1000 | 9990 | 860 | 3.5 | 35.7 | 3.4 | 33.5 |
> | G45 | 1000 | 9990 | 2080 | 3.5 | 35.6 | 3.4 | 33.5 |
> | G46 | 1000 | 9990 | 1340 | 3.5 | 35.6 | 3.4 | 33.5 |
> | G47 | 1000 | 9990 | 2040 | 3.5 | 35.6 | 3.4 | 33.5 |
> | G48 | 3000 | 6000 | 0 | 4.8 | 48.8 | 3.3 | 3.6 |
> | G49 | 3000 | 6000 | 0 | 4.8 | 48.7 | 3.6 | 2.6 |
> | G5 | 800 | 19176 | 280 | 4.1 | 41.9 | 5.2 | 51.9 |
> | G50 | 3000 | 6000 | 3380 | 4.8 | 48.7 | 4.2 | 42.1 |
> | G51 | 1000 | 5909 | 1620 | 3.1 | 32.3 | 2.5 | 24.8 |
> | G52 | 1000 | 5916 | 1560 | 3.1 | 32.2 | 2.5 | 24.8 |
> | G53 | 1000 | 5914 | 2340 | 3.1 | 32.2 | 2.5 | 24.8 |
> | G54 | 1000 | 5916 | 2620 | 3.1 | 32.3 | 2.5 | 24.8 |
> | G55 | 5000 | 12498 | 16840 | 7.1 | 71.5 | 7.3 | 73.1 |
> | G58 | 5000 | 29570 | 27080 | 8.6 | 84.6 | 11.2 | 111.7 |
> | G60 | 7000 | 17148 | 56440 | 9.2 | 91.5 | 10.1 | 100.8 |
> | G63 | 7000 | 41459 | 126360 | 11.2 | 111.0 | 16.1 | 160.9 |
> | G70 | 10000 | 9999 | 227300 | 11.2 | 109.7 | 11.1 | 111.3 |
>
>
>
> [1] Benlic, Una, and Jin-Kao Hao. "Breakout local search for the max-cutproblem." Engineering Applications of Artificial Intelligence 26, no. 3 (2013): 1162-1173.

---

### Official Review · Reviewer_sKQy · 2024-11-02

**Soundness:** 2
**Presentation:** 2
**Contribution:** 2
**Rating:** 6
**Confidence:** 4

**Summary:**

In this paper, the authors propose a framework that combines exploration and exploitation for combinatorial optimization (CO). The proposed framework explores the search space by generating a pool of solutions and exploits the promising ones through refinement. The model is based on Graph Isomorphism and Graph Attention Network, it outputs soft solutions that are heuristicly converted to hard solutions. The framework is applied and tested on three graph CO problems: the Maximum Clique Problem, the Maximum Independent Set Problem, and the Maximum Cut Problem.

**Strengths:**

- The idea of using K-coupled solutions for exploration and Iterative Stochastic Refinement for exploitation looks original and promising.
- The model shows excellent results; the proposed method outperforms state-of-the-art learning-based approaches not only on the training distribution but also in terms of generalization to larger problem sizes.

**Weaknesses:**

- Main concern: reproducibility seems impossible. There are no details about the implementation, only a brief description of the architecture  ('2L layers' of GIN and GAT), with no further details. There is no mention at all of the hyperparameters and the training process.
- The proposed framework is tailored to a small subclass of CO problems. It can be applied to simple graphs, defined by their adjacency matrices, and to problems where solutions can be represented as binary decisions for each node. This makes the framework inapplicable to other classes of CO problems, such as routing or scheduling, as well as to any graph problems with node or edge features.
- Although the paper claims that the method promotes solution diversity by generating multiple solutions simultaneously, in practice, the pool contains only two solutions. The method struggles when more diverse solutions are provided.

**Questions:**

1. Is it possible to apply this approach to other combinatorial optimization (CO) problems, such as routing or scheduling? Or to graphs with node/edge features (e.g., the Maximum Weighted Independent Set/Clique)?

2. What is the motivation for using Graph Isomorphism Networks (GIN) and Graph Attention Networks (GAT) and constructing the multilayer graph in the way described? There is no theoretical or empirical discussion justifying this choice. The ablation study shows that using more than two coupled layers degrades performance, which is really surprising This may suggest that GAT struggles to propagate information effectively across more than two solutions and/or that the proposed simple multilayer graph, which connects only copies of the same node in G, is not powerful enough to represent relations between solutions. Did you try using a more sophisticated multilayer graph and/or a different method to aggregate the data between solutions?

3. Following this, the multilayer graph for 2-coupled solutions (as used in the experiments) is very simple - it has 2N nodes and N edges (one edge per pair of corresponding original nodes). GAT is designed to aggregate information from many neighboring nodes, so using it on such a simple graph (in effect, it computes attention between just two nodes) seems odd and possibly unnecessary. Wouldn't a simple MLP achieve the same result?

4. In the discussion of experiments, much emphasis is placed on comparing results based on running time, but no details are provided on how the experiments were conducted. Were the solvers and models run on the same hardware? Were they tested under the same conditions (e.g., serial or parallel execution)? Neural networks can often solve multiple instances in parallel batches on GPUs, which might not be the case for solvers executed on CPUs (which are inherently much slower than GPUs by design). Claims about running times are only comparable if all methods are tested under similar conditions; otherwise, the comparison could be confusing. E.g. claim No. 4 "We additionally allow solvers a 30-minute time limit, which is at least 24 times longer than our longest-running model." could be misleading. By checking results, Gurobi is in most cases much faster than the proposed method (e.g. in Table 1 Gurobi vs. longest-running model for RB250 is 0.31s vs. 1.41s).

5. All CO problems have simple greedy heuristics, such as choosing the node with the smallest degree for the MIS problem. Did you attempt to exploit this for the initialization of node features (e.g., assigning lower probabilities to high-degree nodes since they are less likely to be part of the solution)? This approach might provide a better initialization than random and could lead to faster learning.

---

> ### Author Response · Authors · 2024-11-21
> **Review 3 Part 1**
>
> Thank you for your thorough review and constructive feedback which have greatly improved our work. We address individual points below:
>
> ## 1. Reproducibility
> > Main concern: reproducibility seems impossible. There are no details about the implementation, only a brief description of the architecture ('2L layers' of GIN and GAT), with no further details. There is no mention at all of the hyperparameters and the training process.
>
> Thank you for pointing this out. We plan to open source the code upon publication to foster reproducibility. Furthermore, we added a new section that will provides descriptions of the architecture, hardware, hyperparameters, and training process.
>
> ## 2. Other CO problems
> > The proposed framework is tailored to a small subclass of CO problems. It can be applied to simple graphs, defined by their adjacency matrices, and to problems where solutions can be represented as binary decisions for each node. This makes the framework inapplicable to other classes of CO problems, such as routing or scheduling, as well as to any graph problems with node or edge features.
>
> Thank you for this comment, we agree that the field of neural solvers has been focused on a somewhat limited set of simple benchmarks. To address this, we have run additional experiments on more complex Max Clique instances from DIMACS [1] and independent set instances on d-regular graphs and reductions from coding theory [2]. Additionally, based on comments from reviewer aRXC, we ran experiments on GSET [3], a well-established max cut benchmark. Lastly, thanks to your comment, we also ran an evaluation on weighted independent set for RB250 graphs. We note that running on these additional datasets not only showcases our applicability to solve instances and weighted versions of the 3 problem types we consider, but also show that X2GNN can be performant for solving broader problems that can be reduced to the investigated problem classes. This is because some instances from these datasets are the result of reducing from problem instances such as coding theory, steiner triple, set covering, and tiling using hypercubes. We are happy to provide more results on RB and ER graphs for MC and MIS, and BA graphs for MCut in addition to our results on in-distribution performance, generalization to larger instances, and generalization to out-of-distribution problems. Overall, X2GNN is flexible enough to handle problems where a feasible solution can be described by decision variables on edges or nodes, and can handle cases where there is problem data on edges or nodes such as weights. One bottleneck here to note is that X2GNN currently operates using an unsupervised loss for combinatorial optimization problems, which is an active area of research.  Nevertheless, it is promising that it can empirically solve other NP-Complete problems after being reduced to the 3 problems we investigate. We will clarify this hint towards broader applicability in the paper.
>
> ## 3. Encouraging Solution Diversity
> > Although the paper claims that the method promotes solution diversity by generating multiple solutions simultaneously, in practice, the pool contains only two solutions. The method struggles when more diverse solutions are provided.
>
> X2GNN encourages solution diversity in multiple ways, penalizing pairs of solutions that are too similar, but also adding randomness globally. We will add a discussion explaining that encouraging diversity between more than two solutions may not yield high-quality solutions as certain critical nodes may be consistently in good solutions and requiring the model to output more unique but still high-quality solutions will degrade performance. Potentially other diversity losses may avoid such an issue and would be interesting avenues for future work.

---

> ### Author Response · Authors · 2024-11-21
> **Review 3 Part 2**
>
> ## Questions
> 1. > Is it possible to apply this approach to other combinatorial optimization (CO) problems, such as routing or scheduling? Or to graphs with node/edge features (e.g., the Maximum Weighted Independent Set/Clique)?
>
> We address this earlier in **Other CO Problems** but simply put, we evaluate on new benchmarks including weighted independent set, and datasets including reductions from other combinatorial problems showcasing the potential for broader applicability.
>
> 2. > What is the motivation for using Graph Isomorphism Networks (GIN) and Graph Attention Networks (GAT) and constructing the multilayer graph in the way described? There is no theoretical or empirical discussion justifying this choice. The ablation study shows that using more than two coupled layers degrades performance, which is really surprising This may suggest that GAT struggles to propagate information effectively across more than two solutions and/or that the proposed simple multilayer graph, which connects only copies of the same node in G, is not powerful enough to represent relations between solutions. Did you try using a more sophisticated multilayer graph and/or a different method to aggregate the data between solutions?
>
>
> This is a good question on using GIN and GAT for encoding the multilayer graph. We use GIN to keep the approach as close as possible to the baselines, most of which also use GIN. The impact of multiple solutions is a good point. We believe performance degradation may be due to the diversity loss we choose rather than GAT layer. Since our diversity loss function encourages solutions to be pair-wise different, increasing the layer count above 2 encourages all solutions to be unique. However, for most instances, there are certain nodes that are part of every good solution. Due to this reason, pushing solutions to be diverse over K=2 forces them to degrade each solution's quality to satisfy diversity. A more sophisticated diversity loss could potentially solve this problem.
>
>
> 3. > Following this, the multilayer graph for 2-coupled solutions (as used in the experiments) is very simple - it has 2N nodes and N edges (one edge per pair of corresponding original nodes). GAT is designed to aggregate information from many neighboring nodes, so using it on such a simple graph (in effect, it computes attention between just two nodes) seems odd and possibly unnecessary. Wouldn't a simple MLP achieve the same result?
>
> We initially experimented using GIN instead of GAT, but GAT performed better. We also now evaluated GAT with a simple MLP as suggested by the reviewer, adding the results to Table 10 in the appendix. The MLP version is still quite performant compared to the baselines; however, is outperformed by the GAT version. We believe that the GAT layer is able to route information over solutions. Each node contains information from its neighbors not just its state, so we believe this is the reason why GAT is more suitable than MLP.

---

> ### Author Response · Authors · 2024-11-21
> **Review 3 Part 3**
>
> 4. > In the discussion of experiments, much emphasis is placed on comparing results based on running time, but no details are provided on how the experiments were conducted. Were the solvers and models run on the same hardware? Were they tested under the same conditions (e.g., serial or parallel execution)? Neural networks can often solve multiple instances in parallel batches on GPUs, which might not be the case for solvers executed on CPUs (which are inherently much slower than GPUs by design). Claims about running times are only comparable if all methods are tested under similar conditions; otherwise, the comparison could be confusing. E.g. claim No. 4 "We additionally allow solvers a 30-minute time limit, which is at least 24 times longer than our longest-running model." could be misleading. By checking results, Gurobi is in most cases much faster than the proposed method (e.g. in Table 1 Gurobi vs. longest-running model for RB250 is 0.31s vs. 1.41s).
>
> The hardware usage distinction is a valid point. We added a new subsection to the appendix for clarification. We run all algorithms on identical machines. Gurobi is run with 4 cores for each instance and other traditional heuristics are run with a single core since they do not directly support parallelization. Since X2GNN creates many K-coupled solutions, it can leverage the parallel nature of the GPUs even when solving an instance at a time. We also want to make clear that Gurobi is able to stop early if it proves optimality. For those small instances, it can solve them quickly so it does not run for 30 minutes in most instances. The motivation for running Gurobi in two settings, with a 30 minute limit, and two much smaller limits, is that we want to evaluate Gurobi in different use-cases. Specifically, Gurobi 30 minute is supposed to allow it as much time as we can afford to let it solve the problem instances. Whereas, Gurobi fast and quality are supposed to evaluate Gurobi’s performance at providing solutions in a comparable runtime to the neural solvers. We will emphasize the actual runtime incurred by Gurobi is not the time limit but rather reported in the table.
>
> 5. > All CO problems have simple greedy heuristics, such as choosing the node with the smallest degree for the MIS problem. Did you attempt to exploit this for the initialization of node features (e.g., assigning lower probabilities to high-degree nodes since they are less likely to be part of the solution)? This approach might provide a better initialization than random and could lead to faster learning.
>
> This is a great suggestion, we refrained from leveraging too many problem-specific heuristics for the various approaches to tease out the algorithmic benefit from using one over the other and keep it more general. However, it would be interesting to see how far we could incorporate any of the learning-based approaches with traditional approaches such as initializing the neural solvers with heuristic solutions (obtained through greedy or from solver internals), using problem-specific local search to improve solutions after they are generated, and even warm starting exact solvers with generated solutions. We are interested in pursuing such avenues in future work.
>
>
> ## References
> [1] https://iridia.ulb.ac.be/~fmascia/maximum_clique/DIMACS-benchmark
>
> [2] https://oeis.org/A265032/a265032.html
>
> [3] GSET- http://web.stanford.edu/~yyye/yyye/Gset/

---

> ### Comment · Reviewer_sKQy · 2024-11-24
>
> I appreciate the authors' detailed response, which addressed all of my questions and comments. However, I remain concerned about the limitation of the proposed method to only a small class of combinatorial optimization problems. While I appreciate the addition of more benchmarks, they are still centered around the same three problems.
>
> I am pleased to see that a small modification allows the proposed method to handle weighted variants of these problems. However, it is difficult to assess the real performance of X2GNN based on the provided experiments in Table 9. The problem size is quite small, Gurobi can solve them very quickly, and no comparisons with other NCO baselines are included.
>
> Additional ablations and reproducibility details improve the quality of the paper, and I will increase my score to 6.

---

### Official Review · Reviewer_m7s4 · 2024-11-03

**Soundness:** 3
**Presentation:** 3
**Contribution:** 3
**Rating:** 6
**Confidence:** 5

**Summary:**

This paper presents an explore-and-exploit Graph Neural Network (GNN) framework for combinatorial optimization (CO) problems. The key idea involves generating multiple solutions simultaneously to facilitate exploration while employing neural stochastic iterative refinement for exploitation. This approach effectively balances exploration and exploitation, leading to high-quality performance. Experiments conducted on three CO problems—namely, the maximum independent set, maximum clique, and maximum cut—demonstrate that the proposed algorithm outperforms learning-based algorithms in the literature.

**Strengths:**

1. The proposed framework utilizes Graph Neural Networks and unsupervised learning to effectively balance exploration and exploitation for combinatorial optimization problems.

2. Empirical results demonstrate high-quality optimization performance and goode generalization capabilities compared to learning-based baselines.

**Weaknesses:**

1. The encoder uses multiple original graphs as input; however, the rationale for connecting identical vertices across these graphs with edges is unclear. Table 5 only presents comparison results for the MIS and MC problems, why are the results for MCut not included? The description of the Drop Value is unclear, it would be better to provide a more detailed comparison of the results. Additionally, the drop value for K=2 shows a significant difference only in the context of MIS.

2. There is a lack of an ablation study on the design of the total loss function. The total loss function includes includes objective quality, constraint satisfaction, and solution diversity. It would be useful to analyze the results when the loss function includes only one of these components, such as solely objective quality, as well as the combination of objective quality and constraint satisfaction, and the combination of objective quality and solution diversity. This comparative analysis could provide insights into the impact of each loss component on the overall performance.

3. The result comparisons for each CO problem contain too few types of benchmarks. MC and MIS are closely related problems, and the instances tested in the experiments should remain consistent. Additionally, it would be beneficial to include more results for RB graphs and ER graphs. For the Max Cut problem, providing more results for BA graphs would also be helpful. Furthermore, testing the proposed algorithm on DIMACS and COLOR02 instances would demonstrate its generalization capabilities.

4. It would be beneficial to explicitly state the limitations of the proposed approach, for example, the scalability issues.

**Questions:**

1. During the iterative refinement process, do the local optimal solutions occur at intermediate steps, or do they only manifest in the final iteration?
2. How are the values ​​of C and T determined for each CO problem?

---

> ### Author Response · Authors · 2024-11-21
> **Review 2 Part 1**
>
> Thank you for your detailed review and helpful suggestions. We address individual points below:
>
> ## 1. Multilayer graph motivation, table 5 comparison, drop value
> > The encoder uses multiple original graphs as input; however, the rationale for connecting identical vertices across these graphs with edges is unclear. Table 5 only presents comparison results for the MIS and MC problems, why are the results for MCut not included? The description of the Drop Value is unclear, it would be better to provide a more detailed comparison of the results. Additionally, the drop value for K=2 shows a significant difference only in the context of MIS.
>
>
> * We will clarify that the multilayer graph is essential for allowing the network to pass information between the generated solutions, giving X2GNN better leverage to encourage diversity. Allowing messages to pass between corresponding nodes allows information to propagate between the two networks without incurring too much overhead in computational complexity.
>
> * For table 5, originally we didn’t have the ablation study as the approach was obtaining near-best solutions for many hyperparameter settings, note that even at the smallest number of iterations, X2GNN was giving less than 0.15% drop value. However, we will include that result not only to show the ablation but also to highlight the performance of X2GNN on max cut.
>
> * We will clarify that drop value is specifically 100 * (1 - the average solution quality divided by the average quality of the virtual best solutions) %. It is a metric used in the previous work [1,2,3].
>
> * The drop value doesn’t benefit as much from further iterations in other settings as X2GNN quickly solves the problem.
>
>
> ## 2. Ablating the loss function
> > There is a lack of an ablation study on the design of the total loss function. The total loss function includes includes objective quality, constraint satisfaction, and solution diversity. It would be useful to analyze the results when the loss function includes only one of these components, such as solely objective quality, as well as the combination of objective quality and constraint satisfaction, and the combination of objective quality and solution diversity. This comparative analysis could provide insights into the impact of each loss component on the overall performance.
>
>
>
> Thank you for pointing this out, it is an interesting question that gets at the heart of our approach since X2GNN requires both the objective and constraints otherwise, the generated solutions are degenerate. The only component that is not strictly necessary is the diversity loss, on which we performed an ablation study in the paper, finding that it consistently improves solution quality. To see intuitively why the objective and constraint losses are necessary, one can imagine that solving max independent set with only an objective function and no constraints imposing independence will simply select all nodes. Furthermore, only using our objective loss will yield such a solution. Similarly, finding a solution satisfying the independence constraints but ignoring the objective yields a trivial empty solution. Again, here, our constraint loss, which penalizes edges that have both incident nodes selected, will push solutions to be empty. Ultimately, both the objective and constraint loss are necessary to obtain anything resembling a reasonable "non-degenerate" solution.

---

> ### Author Response · Authors · 2024-11-21
> **Review 2 Part 2**
>
> ## 3. Additional results on broader problems
> > The result comparisons for each CO problem contain too few types of benchmarks. MC and MIS are closely related problems, and the instances tested in the experiments should remain consistent. Additionally, it would be beneficial to include more results for RB graphs and ER graphs. For the Max Cut problem, providing more results for BA graphs would also be helpful. Furthermore, testing the proposed algorithm on DIMACS and COLOR02 instances would demonstrate its generalization capabilities.
>
>
> For benchmark problems, we consider instances following previous work developing neural primal heuristics. We agree that the field could use a broader set of benchmark instances. To promote this, we have run additional experiments on Max Clique instances from DIMACS [4] and independent set instances on d-regular graphs and reductions from coding theory [5]. Additionally, based on comments from reviewer aRXC, we ran experiments on GSET [6], a well-established max cut benchmark. Lastly, we also ran an evaluation on weighted independent set for RB250 graphs. We note that running on these additional datasets not only showcase our applicability to solve instances from the 3 problem types we consider, but also show that X2GNN can be performant for solving broader problems that can be reduced to the investigated problem classes. This is because some instances from these datasets are the result of reducing from problem instances such as coding theory, steiner triple, set covering, and tiling using hypercubes. We are happy to provide more results on RB and ER graphs for MC and MIS, and BA graphs for MCut in addition to our results on in-distribution performance, generalization to larger instances, and generalization to out-of-distribution problems.
>
> ## 4. Limitations
> > It would be beneficial to explicitly state the limitations of the proposed approach, for example, the scalability issues.
>
>
> This is a good point, we will explicitly state the limitations in terms of scalability to very dense instances for max cut and independent set, and sparse instances for max clique. We also want to clarify that X2GNN can provide high-quality independent set solutions on sparse graphs with millions of nodes. Additionally, we will clarify that X2GNN as a primal heuristic lacks optimality guarantees.
>
>
> ## Questions
>
> 1. > During the iterative refinement process, do the local optimal solutions occur at intermediate steps, or do they only manifest in the final iteration?
>
> We will clarify that during iterative refinement, the locat optimal solutions occur at **intermediate steps**.
>
> ---
>
> 2. > How are the values ​​of C and T determined for each CO problem?
>
> We determined C and T empirically, noting that C and T do not impact the model training, meaning that they can be adaptively determined based on a pre-trained model. While we do not specifically optimize them using approaches like binary search or grid search, in practice this may be useful to tune C and T according to the problem at hand. Additionally, we will clarify that C*T provides a good estimate of the computational burden as it corresponds to the number of solutions generated. This can be helpful when determining C and T based on the specific practitioner use case. Furthermore, we find that C encourages exploration and T encourages exploitation which can help guide their selection.
>
>
> ## References
> [1] Sun, Zhiqing, and Yiming Yang. "Difusco: Graph-based diffusion solvers for combinatorial optimization." Advances in Neural Information Processing Systems 36 (2023): 3706-3731.
>
> [2] Zhang, Dinghuai, Hanjun Dai, Nikolay Malkin, Aaron C. Courville, Yoshua Bengio, and Ling Pan. "Let the flows tell: Solving graph combinatorial problems with gflownets." Advances in neural information processing systems 36 (2023): 11952-11969.
>
> [3] Li, Yang, Jinpei Guo, Runzhong Wang, and Junchi Yan. "From distribution learning in training to gradient search in testing for combinatorial optimization." Advances in Neural Information Processing Systems 36 (2024).
>
> [4] https://iridia.ulb.ac.be/~fmascia/maximum_clique/DIMACS-benchmark
>
> [5] https://oeis.org/A265032/a265032.html
>
> [6] GSET- http://web.stanford.edu/~yyye/yyye/Gset/

---

> ### Comment · Area_Chair_PbN1 · 2024-11-25
>
> Dear Reviewer,
>
> This is a kind reminder that the dicussion phase will be ending soon on November 26th. Please read the author's responses and engage in a constructive discussion with the authors.
>
> Thank you for your time and cooperation.
>
> Best,
>
> Area Chair

---

### Official Review · Reviewer_Kcjm · 2024-11-12

**Soundness:** 3
**Presentation:** 3
**Contribution:** 2
**Rating:** 6
**Confidence:** 4

**Summary:**

The paper proposes an iterative neural approach to search for solutions to graph combinatorial optimization (CO) problems. The approach is based on two phases: the generation of a diverse pool of solutions and their iterative improvement. The training is unsupervised and relies on a composite loss that combines the continuous relaxation of the CO problem objective, a penalization of the constraint violation and a diversity-encouraging term. The approach is evaluated on three graph CO problems and shows a very good performance compared to learning and non-learning based methods as well as a strong generalization performance.

**Strengths:**

* Novel unsupervised framework to generate solutions to graph CO problems
* The framework components, in particular the architecture and the loss are generic and should apply to a variety of CO problems defined on unweighted graphs.
* Original way to deal with several solutions for a given instance by constructing a "K-coupled graph" that allows to capture the whole collection of solutions to input to the refinement step.
* Strong and consistent performance on the three problems and nice generalization to larger instances
* The paper cites and compares to a number of relevant baselines in the experiments

**Weaknesses:**

* While the paper explains well how the hyperparameters (K, C, T, $\phi$) control the exploration/exploitation trade-off, the paper does not provide clear guidelines on how to choose good values for these hyperparmeters, except a grid-search.
    * In Sec 5.5, the paper claims L253 "Different search strategies are needed for MC and MIS due to the different feasible regions. MC requires exploration to avoid local optima, and MIS requires exploitation to improve solutions." I don't understand this argument, can the authors elaborate on this?
    * In general, for all CO problems and search methods, there is a risk of getting trapped in a local minima and a need for solution improvement. I can't see how one can decide beforehand what is more important for a given problem, especially since it may depend on the instances.

* Comparing the run times between learning-based approaches which usually run on GPUs and OR solvers which run on CPUs is always delicate to interpret and gives a partial view of the efficiency of the methods. While there is no straightforward way to make the comparison more fair, it should at least be acknowledged.
   * In addition, the paper does not provide information on the machines on which the experiments were done -- this is especially important to appreciate the claims on the run times.

* The main paper contributions are to compute meaningful output probabilities on the nodes but then only a simple rule or a greedy method is applied to construct a feasible solution (See paragraph Converting Soft Solutions to Hard Solutions).
   * Using a threshold of 0.5 seems arbitrary to chose whether or not a node is part of the solution. Did the author try other values? How one can choose this threshold for a new problem?
   * Given the probabilities, more sophisticated search methods can be applied such as beam search, Monte Carlo tree search or a least stochastic sampling (similarly to what is done when the model outputs heatmaps for example in the cited DIFUSCO method).
   * Evaluating the proposed approach in combination with a stronger search technique, like the above, would be interesting and strengthen the claims.
   * The question being: is the proposed approach useful only when a simple rule is used to construct the solutions or is it also helpful when combined with more sophisticated search?

**Questions:**

* L258: the paper states that the training is done in two stages. Are they done sequentially or alternatively? The arrows in Figure 1 towards the "loss block" made it confusing to me.

* In the Ablation section, when evaluating the impact of K, what was the value of C? In particular, it's important to evaluate the effect of K=1 with a large C, to demonstrate the value of the K-coupled solutions.

* Remarks:
  * L245, L252 it may be misleading to state that the diversity is "imposed" through a loss, "encouraged" would be more clear.
  * It would be helpful to give an explanation of the corresponding equations L249 and L254
  * Since at training, T=1, the authors could get rid of the t index in the description to lighten the notations

---

> ### Author Response · Authors · 2024-11-21
> **Review 1**
>
> Thanks for your detailed review and fruitful comments. We address individual points below:
>
> ## 1. Hyperparamters
> 1. > While the paper explains well how the hyperparameters (K, C, T, $\phi$) control the exploration/exploitation trade-off, the paper does not provide clear guidelines on how to choose good values for these hyperparmeters, except a grid-search.
>
> We determined the hyperparameters empirically, providing empirical validation for the selection of K and ablation of C and T. We note that C and T do not impact the model training, meaning that they can be adaptively determined based on a pre-trained model. While we do not specifically optimize them using approaches like binary search or grid search, in practice this may be useful to tune C and T according to the problem at hand. Additionally, we will clarify that C*T provides a good estimate of the computational burden as it corresponds to the number of solutions generated. This can be helpful when determining C and T based on the specific practitioner use case. Furthermore, we find that C encourages exploration and T encourages exploitation which can help guide their selection. For $\phi$, during training we sampled $\phi$ randomly to make the model robust to various values of $\phi$. Then at inference time, we selected $\phi$ based on validation performance without retraining. We will make this clearer in the paper.
>
> 2. >In Sec 5.5, the paper claims L253 "Different search strategies are needed for MC and MIS due to the different feasible regions. MC requires exploration to avoid local optima, and MIS requires exploitation to improve solutions." I don't understand this argument, can the authors elaborate on this?
>
> That is a good point, and we will clarify the takeaway of this experiment to be that there is an overall phenomenon when solving MC instances that more exploration is required compared to exploitation, and when solving MIS instances, more exploitation is required compared to exploration. We find that X2GNN exhibits good performance with the corresponding set of hyperparameters in each of these settings, giving some support to the idea that these hyperparameters control exploration and exploitation, two high-level concepts. We will better explain that when solving MC instances, any selected node will greatly determine the rest of the nodes in the clique (specifically all adjacent nodes to the initial node). Thus it is difficult to locally improve a max clique solution by making small changes, since the full clique itself is highly determined by any selected node in the clique. Max clique benefits from exploring the search space by identifying many diverse cliques and selecting the best one. Setting the hyperparameters of X2GNN towards exploration and away from exploitation (selecting a high C value and a low $\phi$ value) gives the best performance in this max clique setting, indicating that the hyperparameters allow for controlling performance in an interpretable manner, rather than just being arbitrary hyperparameters to tune. Similarly, for independent set, an approach may greatly benefit from iteratively refining a solution for instance by making small changes to the selected node set. Similarly, guiding X2GNN towards exploitation improves performance here.
>
> 3. >In general, for all CO problems and search methods, there is a risk of getting trapped in a local minima and a need for solution improvement. I can't see how one can decide beforehand what is more important for a given problem, especially since it may depend on the instances.
>
>
> Ultimately, intuition for solving a problem is somewhat based on the problem at hand. We will clarify that if the domain practitioner has high-level knowledge about the problem at hand, they may leverage it to tune the hyperparameters, or alternatively learn something about the problem at hand based on what hyperparameters are performant. It may be necessary to tune these parameters based on a few problem instances.

---

> ### Author Response · Authors · 2024-11-21
> **Part 2**
>
> ## 2. Hardware-related considerations
>
> 1. > Comparing the run times between learning-based approaches which usually run on GPUs and OR solvers which run on CPUs is always delicate to interpret and gives a partial view of the efficiency of the methods. While there is no straightforward way to make the comparison more fair, it should at least be acknowledged.
>
>
> Thank you for this point, we will be sure to include a discussion of the hardware considerations inherent in our experimental design. We agree that while the comparison between traditional and neural solvers doesn’t use the same hardware, it directly parallels the design decisions and resulting runtime that real-world practitioners would plan around when considering which method to use. In the settings we consider, we assume that we give each method access to the best hardware that it can leverage (and that we can afford). Ultimately, to our knowledge, many traditional solvers like KaMIS and Gurobi cannot leverage GPU resources and cannot be easily adapted to benefit from GPU computation. Furthermore, if GPU resources are available to the practitioner either on-site or through cloud services, it may make more sense to use approaches such as X2GNN, which can significantly accelerate with more powerful GPUs. Additionally, we acknowledge that if the practitioner is constrained to only use CPU, then it is likely the case that neural solvers are inappropriate. We will also reference work to bridge the hardware gap in learning-based combinatorial solvers [1].
>
> 2. > In addition, the paper does not provide information on the machines on which the experiments were done -- this is especially important to appreciate the claims on the run times.
>
> Thank you for this suggestion, we have added a description of the machines we use to run experiments and the resource limitations we impose on individual jobs to section A2 to help practitioners get a better idea of the runtime they might expect to see depending on their hardware.
>
> ## 3. Treatment of node probabilities
> 1. > The main paper contributions are to compute meaningful output probabilities on the nodes but then only a simple rule or a greedy method is applied to construct a feasible solution (See paragraph Converting Soft Solutions to Hard Solutions).
>
>
>
> It is a good observation that the model’s outputted probabilities are not leveraged fully. However, this is broadly due to the fact that the outputted values are empirically quite binary, either 0 or 1, or close to it, and do not represent probabilities. This challenge of calibration, where neural networks output values between 0 and 1 but which do not actually represent probabilities, is an interesting area of research and would be interesting to introduce generally into the space of learning for optimization. We will clarify this empirical finding and discuss the implications for solution decoding procedures.
>
> 2. > Using a threshold of 0.5 seems arbitrary to chose whether or not a node is part of the solution. Did the author try other values? How one can choose this threshold for a new problem?
>
>
> That is a good point that the threshold of 0.5 for max cut may not be the best, and indeed we might consider other thresholds. Taking this to the extreme, we might consider all possible thresholds, effectively ordering the nodes by their scores, and then linearly searching splits, putting everything to the left on one side of the cut and everything to the right on the other side of the cut. We can then select the best searched cut. This decoding method is guaranteed to do no worse than the 0.5 threshold decoding. We note that X2GNN already improves over the investigated baselines on max cut including traditional OR approaches, and this will only benefit our approach. We are actively running that experiment now and plan to present the results soon.
>
> 3. > Given the probabilities, more sophisticated search methods can be applied such as beam search, Monte Carlo tree search or a least stochastic sampling (similarly to what is done when the model outputs heatmaps for example in the cited DIFUSCO method).
>
>
> While beam search or Monte Carlo methods would be interesting if the outputted values do represent calibrated probabilities, they are not as applicable for uncalibrated models. This is the reason why we suggest using a stochastic refinement process that adds noise to randomly sampled nodes. It would be interesting for future work to modify neural solvers to encourage calibration or possibly to incorporate more sophisticated search methods during training but as it stands, the models themselves are trained agnostic to the decoding approach.

---

> ### Author Response · Authors · 2024-11-21
> **Part 3**
>
> 4. > Evaluating the proposed approach in combination with a stronger search technique, like the above, would be interesting and strengthen the claims.
>
>
> It would be interesting future work to understand the impact of stronger search techniques globally for various non-autoregressive neural optimizers, and would also be interesting to investigate how these decoding methods could be integrated into the training procedure to train end-to-end. Another interesting avenue of research is to train a second model to learn which nodes should be noised instead of sampling them uniformly at random.
>
> 5. > The question being: is the proposed approach useful only when a simple rule is used to construct the solutions or is it also helpful when combined with more sophisticated search?
>
>
> Decoding selection is an interesting overall question. Ultimately, a decision-maker would deploy the most performant method that works best in their setting, whether it is simple or sophisticated. Furthermore, it would be interesting to see how X2GNN could be tightly integrated into sophisticated solvers, going from simple greedy decoding, to stochastic sampling, and finally to integrating X2GNN to help guide exact solvers like Gurobi or KaMIS. We believe that in future work, X2GNN’s ability to generate solutions or solution-like objects can have broader applications outside of solution prediction such as selecting branching priority, guiding large neighborhood search, and more. We will modify our writing to include a discussion of alternative search strategies.
>
> ## Questions
> 1. > L258: the paper states that the training is done in two stages. Are they done sequentially or alternatively? The arrows in Figure 1 towards the "loss block" made it confusing to me.
>
> We will better clarify and motivate that the training is done sequentially. In the initial stage, X2GNN is only trained as a solution generator. In the second stage, X2GNN is trained to mimic its full deployment using two iterations of applying the model, and imposing losses on both the generated solutions outputted by the first model application, and the improved solutions outputted by the second model application. As such, in the second stage, the weights are updated from gradients from both the solution refinement, as well as the solution generation, effectively including the initial training phase. We perform two-stage training since, without this, the model learns to mostly rely on refinement and the quality of generated solutions suffers. Two stage training ensures that the model first learns to generate better solutions without relying on iterative refinement.
>
> 2. > In the Ablation section, when evaluating the impact of K, what was the value of C? In particular, it's important to evaluate the effect of K=1 with a large C, to demonstrate the value of the K-coupled solutions.
>
>
> We will make it clearer that C is determined by K so that C*K is a constant, meaning that we have a constant number of solutions and approximately a constant amount of computation. So for K=1, C is large.
>
> ### Remarks
> Thank you for these remarks which we will use to improve the clarity of our approach.
>
> 1. > L245, L252 it may be misleading to state that the diversity is "imposed" through a loss, "encouraged" would be more clear.
>
> You are correct that we are not guaranteeing constraint satisfaction through the loss but rather encouraging it and then enforcing it through our solution decoding.
>
> 2. > It would be helpful to give an explanation of the corresponding equations L249 and L254
>
>
> We will also give explanations and intuition for the diversity equations in that for MC and MIS, we penalize dot product similarity between the nodes selected for pairs of solutions, and for MCut we penalize dot product similarity between the cut edges.
>
> 3. > Since at training, T=1, the authors could get rid of the t index in the description to lighten the notations
>
> Thank you for this suggestion. While in the first stage of training, we use T=1 (training the model for solution generation), in the second stage of training, we have T=2. We will clarify this in the paper. Additionally, during deployment we vary T and the index beyond just 2 to get improved performance.
>
> ## References
>
> [1] Gupta, Prateek, Maxime Gasse, Elias Khalil, Pawan Mudigonda, Andrea Lodi, and Yoshua Bengio. "Hybrid models for learning to branch." Advances in neural information processing systems 33 (2020): 18087-18097.
>
> [2] Guo, Chuan, Geoff Pleiss, Yu Sun, and Kilian Q. Weinberger. "On calibration of modern neural networks." In International conference on machine learning, pp. 1321-1330. PMLR, 2017.

---

> > ### Comment · Reviewer_Kcjm · 2024-12-02
> > **Reply to Authors Rebuttal**
> >
> > I would like to thank the authors for the precise answers to my questions and concerns. I appreciated the clarifications about the outputted scores which are not probabilities and the discussion about the hyperparameters and especially the added experiments with the output threshold. I will increase my score to 6.

---

> ### Comment · Area_Chair_PbN1 · 2024-11-25
>
> Dear Reviewer,
>
> This is a kind reminder that the dicussion phase will be ending soon on November 26th. Please read the author's responses and engage in a constructive discussion with the authors.
>
> Thank you for your time and cooperation.
>
> Best,
>
> Area Chair

---

### Author Response · Authors · 2024-11-21
**Global Rebuttal**

We thank the reviewers for their insightful reviews and feedback which we have used to improve our work. We are happy that the reviewers consider our approach using K-coupled solutions to be novel (Kcjm, m7s4, sKQy, aRXC) and performant, demonstrating generalization to larger instances (Kcjm, m7s4, sKQy). The reviews are filled with good ideas for improvements, and we have added many of the requested experiments which have greatly improved the evaluation of X2GNN.

The major updates we have made in addition to clarification are:
* **Additional benchmarks**: We thank the reviewers (m7s4, sKQy, and aRXC) for prompting us to evaluate X2GNN on additional datasets, and CO problems. We have added empirical evaluation on max clique instances from DIMACS [1] (Appendix Table 7), independent set on d-regular graphs (Appendix Figure 4) and reductions from coding theory [2] (Appendix Table 8), and max cut on GSET [3] (Appendix Table 12). Surprisingly, we found that X2GNN can generalize from training on random graphs to finding optimal, or virtual best solutions when optimality is not yet proven, on many instances from the fixed benchmarks (DIMACS, coding theory, GSET). Additionally, when solving max independent set (MIS) on d-regular graphs, we found that X2GNN is capable of scalably generalizing to instances with three orders of magnitude more nodes, and improves on larger instances in both runtime and solution quality over KaMIS, a specialized MIS heuristic. We also ran evaluation on weighted independent set for RB250 graphs, achieving a drop value of less than 0.8% showcasing the ability to handle weighted instances (Appendix Table 9).
* **Further ablation study**: We add additional experiments aggregating information over solutions using MLP instead of GAT showing that while X2GNN with MLP still outperforms the baselines, it does benefit from using GAT (Appendix Table 10).
* **New baseline**: We thank reviewer aRXC for pointing us to PI-GNN [4]. We evaluate on the same benchmarks as PI-GNN, demonstrating improved solution quality, runtime, and the ability to scale to instances with millions of nodes.
* **Reproducibility**: We have added details about our architecture, selected hyperparameters, training regime, and computing environment. We additionally plan to release our code and trained model weights upon publication.

We are also eager to interact with the reviewers throughout the rebuttal period.

## References
[1] https://iridia.ulb.ac.be/~fmascia/maximum_clique/DIMACS-benchmark

[2] https://oeis.org/A265032/a265032.html

[3] GSET- http://web.stanford.edu/~yyye/yyye/Gset/

[4] Schuetz, M.J., Brubaker, J.K. and Katzgraber, H.G., 2022. Combinatorial optimization with physics-inspired graph neural networks. Nature Machine Intelligence, 4(4), pp.367-377.

---

> ### Author Response · Authors · 2024-11-25
> **General update with new experiments**
>
> We thank the reviewers for their reviews and suggestions which have greatly improved the quality of our paper. We also thank reviewer sKQy for responding positively to our response, additional ablations, and reproducibility efforts. We hope to broaden the scope of NCO to a wider range of problems in future work. We would like to update the reviewers with additional experiments that are relevant to their questions. We are eager to answer any remaining questions and discuss our work throughout the rebuttal period.
>
> ## Reviewer 1 (Kcjm)
> > Using a threshold of 0.5 seems arbitrary to chose whether or not a node is part of the solution. Did the author try other values? How one can choose this threshold for a new problem?
>
>
> We will add that the predicted values for max cut are very close to binary, with 90% of the decision variables being within 1e-5 of the nearest binary value.
>
>
> For thresholding max cut solutions, we note that we can consider all thresholds on the fly as this corresponds to sorting the nodes by output score and doing a linear search over the nodes, putting everything with lower score on one side of the cut, and everything else on the other side of the cut. These cuts can each be evaluated, and we can select the best cut in terms of objective value in polynomial time. We call this approach dynamic thresholding and note that this approach is guaranteed to do no worse in objective value than fixed thresholding since the fixed threshold is one of the cuts evaluated. We perform an experiment on BA250 max cut instances, reporting results in table 13 in the appendix. Dynamic thresholding gives a 0.01% improvement in solution quality for various models of X2GNN at the cost of increased runtime. Overall, we consider that more iterations of iterative refinement may be a better use of limited runtime. We will clarify this in the appendix.
>
>
> ## Reviewer 2 (m7s4)
> > Table 5 only presents comparison results for the MIS and MC problems, why are the results for MCut not included?
>
>
> Thank you for this suggestion, we perform an ablation study on K for max cut and add it to table 5. This experiment shows improved performance for K=2. However, it also highlights that the problems are solved very near to optimality for various values of K. We will make sure to clarify this in the paper.

---

### Meta-Review · Area_Chair_PbN1 · 2024-12-19

**Metareview:**

This paper introduces an unsupervised neural framework ($X^2$GNN) that combines exploration and exploitation for combinatorial search optimization.   $X^2$ GNN iteratively refines (exploit) a solution pool (explore) using GNN, generalizing across problem distribution and size, outperforming ML and traditional OR baselines. Most reviewers agree that this paper is novel and performant. During the rebuttal phase, the authors addressed most of the reviewers' concerns, and I suggest that the authors revise the manuscript accordingly. Overall, I recommend accepting this paper.

**Additional Comments On Reviewer Discussion:**

Reviewers Kcjm, m7s4, sKQy, aRXC rated this paper as 5: borderline reject (raised to 6), 6: borderline accept (keep the score), 5: borderline reject (raised to 6), and 5: borderline reject (keep the score), respectively.  The reviewers raised the following concerns.

- Selection of hyperparameters (rasied by Reviewers Kcjm and aRXC).
- Limited applicability (rasied by Reviewer sKQy).
- Scalability (rasied by Reviewers m7s4 and aRXC).
- Unclear experiment details (rasied by Reviewer Kcjm).
- Insufficient experiments (rasied by Reviewers m7s4 and aRXC ).
- Reproducibility (rasied by Reviewer sKQy).

By additional experiments and more details in the rebuttal, the authors address most concerns about scalability, unclear experiment details, insufficient experiments, and reproducibility. Rreviewer aRXC still has concerns about sensitivity of the parameters in penalization methods. However, the sensitivity of parameters is a common issue in the domain. Despite the above issue, this paper still has its contributions. I strongly encourage the authors to incorporate the dicussions to further address the concern in the final version. Overall, I recommend accepting this paper.

---

### Decision · Program_Chairs · 2025-01-22

Accept (Poster)